**Meteorological observations in tall masts for mapping of atmospheric flow in Norwegian fjords**
Birgitte R. Furevik[1], Hálfdán Ágústsson[2], Anette Lauen Borg[1], Midjiyawa Zakari[1,3], Finn Nyhammer[2] and Magne Gausen[4]
[1]Norwegian Meteorological Institute, Allégaten 70, 5007 Bergen, Norway
[2]Kjeller Vindteknikk, Norconsult AS, Tærudgata 16, 2004 Lillestrøm, Norway
[3]Norwegian University of Science and Technology, Trondheim, Norway
[4]Statens vegvesen, Region Midt, Norway
*Correspondence to*: Birgitte R. Furevik (birgitte.furevik@met.no)
**Abstract.** Since 2014, 11 tall meteorological masts have been erected in coastal areas of mid-Norway in order to provide
observational data for a detailed description of the wind conditions at several potential fjord crossing sites. The planned fjord
crossings are part of the Norwegian Public Roads Administration (NPRA) Coastal Highway E39-project. The meteorological
masts are 50 - 100 m high and located in complex terrain near the shoreline in Halsafjorden, Julsundet and Storfjorden in the
Møre og Romsdal county of Norway. Observations of the three-dimensional wind vector are made at 2-4 levels in each mast,
with a temporal frequency of 10 Hz. The dataset is corroborated with observed profiles of temperature at two masts, as well
as observations of precipitation, atmospheric pressure, relative humidity and dew point at one site. The first masts were
erected in 2014 and the measurement campaign will continue to at least 2024. The current paper describes the observational
setup and observations of key atmospheric parameters are presented and put in context with observations and climatological
data from a nearby reference weather station. The 10-minute and 10 Hz wind data as well as other meteorological
parameters are publicly available through Arctic Data Centre ( DOI: 10.21343/z9n1-qw63; Furevik et al., 2019).
**1 Introduction**
In 2014, the Norwegian Public Roads Administration (NPRA) started an evaluation of the environmental conditions, i.e.
wind, atmospheric turbulence, waves and currents, pertaining to making the E39 road 'ferry-free' between Kristiansand and
Trondheim on the western coast of Norway. If realised, the project will include new crossings of eight of the largest fjords in
Norway. The fjords are typically surrounded by steep mountains going up to 500 m. Fjord widths are 2-7.5 km, and water
depths 200-1300 m. This requires a detailed understanding of the wind, wave and ocean current climate at the proposed
crossings which is achieved partly through a large atmospheric and oceanographic measurement programme.
In mid-Norway new fjord crossings are planned in Vartdalsfjorden, Sulafjorden and Halsafjorden, as well as near Julsundet.
The observational campaign started here in 2014, with a considerable increase in measurement effort in October 2016. The
observational programme will continue for at least 8 years, but may be extended to 12 years or more. The program includes
tall meteorological masts erected and operated by Kjeller Vindteknikk (KVT), equipped with sonic anemometers at several
evelations, observing with a temporal resolution of 10 Hz. The most recent masts are 70-100 m high while the masts erected
first have an elevation of ~50 m. A number of wave buoys with meteorological and oceanographic measurements have also
been installed. Similar measurement campaigns are carried out by the NPRA in other fjords, such as Bjørnafjorden, but these
data are not publicly available. The fjord measurement programme of the NPRA is unique in Norway, both in terms of
measurement density, parameters measured and the time frame. To the authors knowledge, there has been no other
dedicated measurement campaign, providing simultaneous and detailed measurements of both the ocean and the lower
atmospheric boundary layer in the complex coastal terrain of western Norway. Oceanographic and atmospheric
measurements have typically been carried out independently and during shorter periods, related to e.g. research programmes
or industry projects. Ongoing large observation programmes include the LoVE, Lofoten–Vesterålen Cabled Observatory of

the shelf marine ecosystem (Godø et al., 2014), the Integrated Carbon Observation System (ICOS)-Norway and Ocean Thematic Centre (OTC) which is an international observation programme of greenhouse gasses (Steinhoff et al. 2019), and the Nansen Legacy, a national research programme which includes extensive observations in the northern Barents Sea and Arctic Ocean (Reigstad et al., 2019). The Norwegian Meteorological Institute (MET Norway) operates a national network of meteorological stations (observational data typically freely available) in the region of the E39 campaign. The NPRA and the National Coastal Administration (NCA) operate meteorological stations in connection with infrastructure and road safety/operations, but wind measurements from these stations may be strongly affected by obstacles and local terrain features. The Frøya meteorological mast recorded ocean wind conditions to form the basis for the NORSOK standard (Andersen and Løvseth, 1995, 2006; Standard Norge, 2017).

From a scientific standpoint, the measurement campaign provides an excellent platform to study the multi-scale variability in boundary-layer flow in complex terrain, and the variation of local flow with regard to the synoptic flow aloft, as previously studied by Jonassen et al. (2011) for southern Norway. The current campaign has already provided unique observations of extreme winds and storms in complex terrain, but here the relevant topographic forcing is typically at a smaller scale than has been studied in many large field campaigns in and near the North Atlantic (e.g. the Norwegian IPY-Thorpex (Kristjánsson et al., 2011) and the Greenland Flow Distortion Experiment (Renfrew et al., 2008)). The boundary layer flow in this part of Norway is to a first order governed by a large scale orographic forcing on the mesoscale and synoptic flow, i.e. due to the high mountains of southern and western Norway. The boundary layer flow may decouple from the flow aloft while the local variability near the surface occurs on scales on the order of a few kilometres, as the flow is for example accelerated along steep mountain slopes and narrow fjords, or stagnates in blocked flow in deep valleys, i.e. in terrain typical for the locations of the masts in the campaign. From a more pragmatic and engineering point of view, the data collection is important for describing the atmospheric forces, both climatic and short-term, pertaining to the technical design of large structures in complex terrain. The data collection is unique in both the length as well as in the detail of the observed time series at the available sites. The series are long enough so that they can be of use in constructing a description of the climatic conditions at the sites, but they are also detailed enough to describe well single weather events of interest and capture some of the complexity in the flow structure on either side of the planned crossings. The wind and turbulence data has already highlighted that for such large structures as are planned, the spatial variability in the flow must be properly accounted for and described.

The objective of the present paper is to provide documentation of the atmospheric part of the E39 dataset and the data handling process for the mast data. The measurement programme is ongoing and the description given here is valid at the time of publication. The paper is structured as follows. Section 2 describes the setup of the observation system, including mast details, the data quality control and an overview of data availability. Access to the data is open, and handled through a new procedure at MET Norway, which is described in section 3. Section 4 presents observations of selected variables to illustrate available parameters and the data quality, and puts the data in context with the long-term regional conditions. A summary is given in section 5.

## 2 Setup of the observation system

As of December 2019, the observational dataset includes observations from 12 tall masts in three main regions of interest in Møre and Romsdal county in western Norway (Fig. 1). All the masts are operational except two: one has been dismantled, and other was discontinued and extended to twice the original height, becoming the 12th mast. Most masts are expected to be operational for at least 8 years, with more details on their setup given below. The masts are located in a region characterized with a relatively complex orography, e.g. narrow and deep fjords, surrounded by steep and high mountains. The conditions are more challenging in the Storfjorden region (region S in Fig. 1), where the brunt of the campaign is focussed, than in the Julsundet and Halsafjorden regions (J and H in Fig. 1). Further details on the setup and conditions at individual masts is given below. Long-term reference surface weather stations, operated by MET Norway, are found within approx. 20 km of each of the main region of interests. Two of these are located in flat terrain at airports, while the third is a located on the small island of Ona. The nearest upper air observations are made ~180 km to the northeast from Ona, at Ørlandet airport.

The main focus of the measurement campaign is to collect data describing the atmospheric and oceanic conditions at possible fjord crossings, pertaining to the dimensioning and design of long structures (suspension bridges and floating bridges have been considered, as well as submerged tunnels). In this aspect, wind is the most important atmospheric variable. The main parameters of relevance can be split into two sets: a) mean quantities which can be described by e.g. the 10-minute mean wind, i.e. the wind speed and direction distribution, return levels of extreme winds and the vertical wind profile. b) turbulence quantities which must be described using observations with a high temporal frequency, e.g. turbulence intensities, the spectral energy density and coherent variations of the turbulence at two locations separated by short distance. Furthermore, the measurement campaign is corroborated by observations from buoys and LIDARs (not presented here), as well long datasets with high-resolution simulations of weather with mesoscale numerical weather prediction and computational fluid dynamics (CFD) models (not presented here).

## 2.1 Masts and instrumentation

A summary of the key parameters for the masts are presented in Table 1, including geographical position, measurement period, base level height and measurement heights. The masts are built and operated by KVT for the NPRA. Observations of wind are made at 2 - 4 levels in each of the masts, while additional atmospheric variables are observed at three sites. Observations are ongoing at all masts, except at Midsund which was dismantled in March 2019, and at Åkvik which became a new station (Åkvik2) in May 2020 with a lenghened mast. The masts are guyed lattice towers (Storfjorden) and tubular masts (Julsundet and Halsafjorden), except at Kårsteinen, Langeneset and Nautneset which are self-supporting lattice masts. Nautneset has previously been instrumented with an accelerometer to verify that the swinging motion of the self-supporting masts has a negligible impact on the intended use of the wind measurements (Tallhaug, 2017).

The three wind components are recorded using three-axial ultrasonic anemometers (Gill WindMaster Pro) which is logging at 20 Hz. The data is subsequently averaged to a temporal resolution of 10 Hz to reduce aliasing. The anemometers are located on 2 - 6 m long horizontal booms, with the boom directions approximately perpendicular to the prevailing and most relevant wind directions (derived a-priori from mesoscale simulations of wind). The true boom direction, as seen from the mast, is presented in Table 1 (average for all levels). The lowermost sensors at the masts at Julbø, Halsaneset and Midsund are located at ~13 m above ground level and have been found to be too strongly affected by their vicinity to the tree top level. This is to some degree also the case for the lowermost sensor at Åkvik (17 magl). In July 2018 a software bug was documented, affecting the vertical wind component of instrument produced before October 2015 (Gill Instruments, 2016). This error has been accounted for and only corrected data are made available as a part of the current dataset. 10 Hz temperature measurements were stored from some of the sonic anemometers, but are not part of the available dataset.

The 10-minute mean wind data is produced from the 10 Hz wind recordings and more than 99.95 % of the 10-minute samples are based on a 50 % or better 10 Hz availability. A 90 % availability of 10 Hz data is found in 99 - 100 % of the 10-minute samples, depending on station. If a 99 % 10 Hz availability is required then the numbers are 96 - 99 % for the 10-minute means. The total uptime for 10-minute mean wind for all sensors and all masts is 98.9%. Instrument failures are fixed at the earliest convenience, with highest priority given to having operational sensors at the top of the masts. An intermittent reduction in 10 Hz data availability is typically associated with errors due to precipitation and other intermittent external or technical disturbances. A malfunctioning instrument or logger will either lead to complete data loss or have sustained periods with a availability far below 100 % for the 10 Hz observations.

The stations Kvitneset_Temp and KvitnesetKlima are located in the same masts as Kvitneset. Kvitneset_Temp has inter-calibrated temperature sensors (PT100 from Campbell Scientific) at the same levels as the wind sensors, with a sampling rate of 0.2 Hz. KvitnesetKlima has measurements (1 Hz sampling rate) of temperature, dew point temperature, relative humidity and air pressure at 9 m above ground level (not corrected to mean sea level). Inter-calibrated temperature measurements at 0.2 Hz (similar as at Kvitneset_Temp) are also done at four levels in the Trælbodneset mast, i.e. at the three levels with wind sensors as well as at 3 m above ground level (here named as Trælbodneset_temp). A Geonor T-200B precipitation gauge is

installed at Brandal (cf. Fig. 2).

## Storfjorden

Storfjorden is the name of the fjord system, which is divided into Sulafjorden, Hjørundfjorden and Vartdalsfjorden in
addition to several other extensions further inland (Fig. 2). Sulafjorden is located approximately 10 km southwest from
Ålesund between the islands Hareidlandet in the west and Sula in the east. The fjord is aligned along a south-southeast north-
northwest axis, and it is ~12 km long from the mainland to the island Godøy and 3-6 km wide. Hareidlandet and Sula have
steep mountains and their upper levels have an elevation of 500 – 700 m asl.  In the south, Sulafjorden connects to
Vartdalsfjorden, a long narrow fjord, which runs perpendicular to Sulafjorden, southwest to northeast. South of
Vartdalsfjorden is Ørsta municipality with Sunnmørsalpene, a high and steep mountain region reaching more than 1200
masl. In the northeast, the narrow Hjørundfjorden connects to Storfjorden, running in a southeast - northwest direction. Fig.
3 provides terrain profiles at all of the masts while Fig. 4 shows a photograph of Sulafjorden at the location of Kvitneset and
Trælbodneset. The largest effort in the measurement campaign of the Coastal Highway E39 project in mid-Norway can be
found here. An overview of the specific conditions at each mast is given below while details were presented in Haslerud
(2019) and references therein.

## Sulafjorden

A precipitation station and four tall meteorological masts are located in Sulafjorden. The masts are located near both ends of
two possible fjord crossing locations. Kvitneset and Langeneset on the western side and Trælbodneset and Kårsteinen on the
eastern side.
The mast at Kvitneset is located on the headland Kvitneset on the northeast corner of Hareidlandet. The headland is a 300 m
wide and 200 m long relatively flat area just below steep mountains reaching up more than 500 m over a distance of 1 km in
the southwest. Fig. 3 shows the terrain profile along a section through the locations at Kvitneset and Trælbodneset, and
serves to highlight the steepness and height of the surrounding mountains.  The masts are located at 6 m asl, in a location
open to the Norwegian Sea in the sector west-northwest to north-northwest. The 10-minute wind data availability is near 100
% for all sensors. There was sporadic loss of 10 Hz data before July 2017 and in March 2019 due to technical issues. The
data availability for the other atmospheric variables is near 100% until December 2018 when it is 0.1-0.9% lower.
A precipitation station was put in operation in March 2018, in the village Brandal between Kvitneset and Langeneset. Due to
a fault, precipitation was not registered during the last 10 days of August 2018.
The Langeneset mast is located to the south in Sulafjorden (i.e. inward) from the mast at Kvitneset. It is mounted in a 100 m
wide industrial area, below a steep mountain side (cf. Fig. 3). The slope is partly covered with an open forest and there are
low buildings in the industrial area. Due to sporadic losses and mast downtime in the summer of 2017 data acquisition
during the first year was 94.6%. For 2018 and onwards the data availability is close to 100 %.
The mast at Trælbodneset is located at 12 m asl, on a small headland on the western side of the island Sula, with view to the
open sea towards the westnorthwest. Towards the east, a mountain rises 450 m over a distance of 1 km (Fig. 3).  The
vegetation is relatively sparse at the mast and along the coast, while the mountainside has open forest.  The 10-minute
availability is 99 - 100% but the top sensor had a slightly later start than the other sensors (16 January 2018). The overall
availability of 10 Hz data is good, with a somewhat reduced availability during some winter months. The 10-minute
availability of the temperature sensors in the masts is near 100 % the first two years, then 92.1 and 97.2 % in 2019 and 2020.
The mast at Kårsteinen is also located on a small headland with a steep mountain rising to 660 m in the northeastern
quadrant (Fig. 2). The mast is located near the opening of Sulafjorden into Vartdalsfjorden.  Due to defect hardware, the
availability was poor during the first few months of operation, but it is near 100% after February 2018. The availability of
10 Hz data is generally good, but relatively low in September 2018.

## Vartdalsfjorden

The mast at Rjåneset is located at the tip of a small peninsula, just west of the settlement at Grøvika, on the southeastern shore of Vartdalsfjorden. There is a mountain rising to 1035 m a few kilometers to the east (Fig. 3), with steep mountainsides in the sector from north - northeast to east, and some of them across the fjord. The headland has some trees and the mountainside is forested. There are some low islands a couple of kilometres to the south and southeast. There are steep mountains across the fjord to the north and west, while the fjord is more open to the southwest where it meets Rovdefjorden and Voldsfjorden. The availability of 10-minute data from the top-most sensor is close to 100 % for the whole measurement period, while due to hardware issues, some data were lost for all sensors during September - November 2018, and after April 2019. The availability of the 10 Hz raw data is generally good, with sporadic losses during summer and slightly increase in the losses during late autumn for both years (2018 and 2019).

## Hjørundfjorden

The mast at Gjeveneset is relatively low compared to the other masts, and is located at a potential building site for the components of a floating structure. The mast is situated at the inlet of Hjørundfjorden at 3 m asl just by the sea, southwest of Hundeidvik, where the fjord opens up towards the north before meeting Storfjorden (Fig. 2). The mast is facing the fjord in the sector south-southeast over west to north, and the land is fairly open towards northeast with spread buildings within a few hundred metres. In the east, open terrain slopes gently up to 20 m over a distance of 200 m and then more steeply up to above 600 m over a distance of 600 m. On both sides of the fjord, steep mountains raise up to more than 1000 m asl. The headland has areas of trees and the mountain side is covered by forest. Data availability from the mast was just over 90% in 2019 due to a hardware failure in the spring. In 2018 and 2020 the availability was good (100%).

## Julsundet

Julsundet is the sound that connects Molde and Fræna municipality on the southeast side and the island municipalities Midsund and Aukra on the northwest side. Julsundet is approximately 17 km long and runs in a north-south direction. On the south side, the sound opens into Moldefjorden, and on the north side into Harøyfjorden. A bridge in the narrowmost part of the sound has been considered, where the width is 2.5 km and mountains reach up to 500 – 600 m on both sides, as seen in Fig. 5 and Fig. 6. Two masts, Midsund (dismantled in spring 2019) and Nautneset, are placed on the western side and one, Julbø, on the eastern side of Julsundet (Fig. 5). The masts at Midsund and Nautneset are only separated by a horizontal distance of ~100 m and have sensors at the same height over mean sea level as well as the same height over ground level. More details are given in Eriksen (2019), and references therein.

Julbø mast is placed on a low headland reaching fairly far into the sound. The topography on the headland goes up to 8 m while the mast is located at 4 masl. There are a few trees and a small cliff down to the sea on the southwest side. The monthly 10-minute data availability is near 100% except during periods associated with technical failures in May, July, November and December 2014, March and July 2017. The 10 hz data availability is generally good, with greater loss during the previously mentioned months.

The Midsund mast was mounted on the west side of the sound, on the Nautneset headland. The headland is forest covered and reaches roughly 300 m into the sound. The topography at the headland reaches up to 50 m with steep cliffs up from the sea. To the west of the headland the terrain rises steeply to 600 m. The mast was mounted 100 m from the outer headland at 24 m asl. The monthly 10-minute data availability is 99 - 100% and the 10 Hz availability typically high, except during periods associated with technical failures in March and August 2014, May and July 2017, as well as June 2018. The Nautneset mast is placed on the harbour about 100 m east of the location of the Midsund mast. The mast has free sight from north (360°) over east to south (180°). In the west the topography rises steeply to Midsund mast and further towards the mountains. In November 2016 - January 2017 the two topmost sensors were out due to a lightning strike, but the lowermost

sensor operated normally, and in March 2019 a technical failure caused loss of data. Apart from this, the data availability has
been close to 100%.

## Halsafjorden

The Halsafjorden fjord runs in a southeast - northwest direction from Todalen in the south, towards the island Tustna (Fig.
7). The fjord is roughly 2.5 km wide at the planned bridge location. The terrain reaches up to 200 – 500 m asl on both sides
and the sides are covered by forest (Fig. 7 and Fig. 8). A mast is placed at Halsaneset on the western side and another,
Åkvik, is placed on the eastern side of the fjord. More details are given in Eriksen (2019), and references therein.
Halsaneset mast is mounted 10 masl, at tip of the headland Halsaneset which reaches 500 m out into Halsafjorden. There are
two small, forested hills (15 and 40 m) on the headland, while the tip of the headland is more sparsely vegetated. The Åkvik
mast is mounted at 6 masl on the tip of a 200 m wide and 500 m long and forest covered headland, Orneset, on the eastern
side of Halsafjorden. The headland is about 80 m high at the farm Haugen and slopes gradually towards the tip while the
southern side of the headland is steep. The height of the mast at Åkvik was increased to 100 m in May 2020 and at the same
time the station got a new name, Åkvik2, and observations stopped at the original station. Due to the short observation series
at Åkvik2, no observations from the station are presented here. Both the Halsaneset and Åkvik masts have a high annual
data-availability of 99.8-100% for 2016-2020.

## 3 Data handling and quality assurance

Data from the sites is handled as follows. Observational data is transmitted in near-realtime to KVT, with a temporary
backup locally stored in the mast loggers. Data is processed and quality checked on an hourly basis at KVT. As the mast
measurements are ongoing and instruments may need replacing, the filtering process is monitored and improved when the
need arises. Furthermore, the operations of the mast observations are monitored in real-time by an automated system which
warns about delays in observations, malfunctioning instruments, missing data or unphysical observed values.
The operational filtering of the 10 Hz wind data made publicly available is threefold. Unphysical values exceeding the
specifications of anemometers are flagged. Noise and data spikes associated with unphysical jumps in the measurement
values are identified and removed from the dataset using a method similar to median filtering. Locked values, i.e. repeated
and identical measurement values for the three wind components, are removed. Further filtering of the available 10 Hz
dataset is not done, and it is left to the user of the data to employ more stringent filtering routines, as he sees fit and needed
for the intended use of the data. Suggestions on applicable filtering methodologies and additional quality assurance are e.g.
given in Hubbard et al. (2012), with more specific details given in in Capozzi et al. (2020) and Steinacker et al. (2011). After
filtering, the observed wind direction in the 10 Hz data is rotated towards true north, and 10-minute means are produced
from the 10 Hz wind data. There is no minimum on the amount of 10 Hz samples used in producing the 10-minute averages,
but the amount can be deduced by inspection of the available 10 Hz data. For other data than observations of wind, the raw-
data are made available as is, and only a first screening of the data is done, with no additional filtering performed.
Hourly data at the native sampling rate and with 10-minute sampling is written to files (netCDF4-format), and are sent to a
virtual server belonging to MET Norway via sftp, typically on a hourly or daily basis. MET Norway performs an additional
quality check on the data, to track any inconsistencies and delays in the data stream. Data from the masts are published as
open access on "http://thredds.met.no". THREDDS (Thematic Real-time Environmental Distributed Data Services) is
software solution run on web servers that provides metadata and data access for scientific datasets, using a variety of remote
data access protocols such as OPeNDAP (Open-source Project for a Network Data Access Protocol). Due to the high data
amount for the 10 Hz wind data, the 10-minute data are stored separately. Both type of files include wind speed, wind
direction and vertical wind speed. The 10-minute averages of the wind observations are based on 10 Hz data from the
interval preceding the time stamp (i.e. labelled right), while the interval is open on the left side and closed on the right side
(i.e. the end points only includes the observation concurrent with the time stamp).

## 4 Wind  conditions and data overview during observation period

The long term automatic weather station Ona II (MET station number 62480) at the island Ona just off the coast (Fig. 1) is
used as a reference station for the wind and temperature measurements. Ona II is operated by MET Norway and data are
available from the open data API: "frost.met.no". Hourly observations of wind speed and direction are available since 2001
(approximately 18 years of data), and they are used to provide a description of the  current state of the long-term regional
wind conditions. For this reference period, the median wind speed at Ona is 6.6 ms$^{-1}$ which varies from 5.1 ms$^{-1}$ in August up
to 8.7 ms$^{-1}$ in January (Fig. 9). Winds above 30 ms$^{-1}$ have been observed in the autumn and early winter, i.e. from September
to December. Since the fjord crossings are separate projects with different timelines and since permits for mounting the
masts are granted separately, all the masts were erected at different times from 11 February 2014 in Julsundet to 14 March
2018 in Hjørundfjord. A 3-year period from Ona II is chosen to represent the period with fjord measurements (Fig. 10 top
left). When compared to the wind speed distribution for the reference period of 18 years (Fig. 9) we see that the wind has
been somewhat weaker during the the chosen 3 years than during the reference period. The median and 75th percentiles of
wind speed during February, July and November are lower than for the whole full series and there have been no recordings
of wind speed above 30 ms$^{-1}$.
At the 11 stations discussed here (Table 2), the lowest annual median wind speed is found in the inner part of Sulafjorden at
Langeneset (2.95 ms$^{-1}$) and Kårsteinen (2.39 ms$^{-1}$) while median wind speed above 5 ms$^{-1}$ are recorded in Julbø (5.15 ms$^{-1}$),
Kvitneset (5.03 ms$^{-1}$), and Rjåneset (5.04 ms$^{-1}$). Strong winds are most frequent in Julsundet and at Kvitneset in Sulafjorden,
while the highest 99th percentiles are found in the inner part of the fjords (Gjeveneset and Rjåneset) in spite of their lower
measurement heights. This is presumably related to the local topography and how well the sites are exposed to direction
associated with strong winds.  The 99th percentile for the, separately, upwards and downwards, oriented vertical winds,
indicates that the strong vertical gusts are often found at the stations in Sulafjorden as well as at Nautneset, compared to at
the other stations, especially those in Halsafjorden.
The wind speed shows a clear seasonal variation at the Ona reference station and most of the masts, except Trælbodneset,
Kårsteinen, Gjeveneset and Rjåneset (Fig. 9 and Fig. 10). Here, the time series are short, and the statistics are less reliable.
The wind roses for the Ona reference station (Fig. 9 and top left in Fig. 11) show that the directional distribution during the
3 year period is quite typical for the  conditions during the last 18 years, as would be expected at a site where the low-level
flow is strongly affected by both the local terrain as well as the large scale orography of western Norway. They also show
that along the coast, the most frequent, as well as the strongest, winds are from the southwest and the northeast, following
the general orientation of the coast. The synoptic scale flow aloft has a large contribution from the south and the east, as well
as a component from the northwest, but the orographic forcing typically deflects such flow along the large scale orography
(see Barstad and Grønås, 2005, and references therein). The wind roses covering the full observation period until april 2019
(Fig. 11) for the 11 stations indicate flow which is strongly affected by the local terrain. Southerly winds (winds blowing
towards the sea) are frequent at all stations, and dominant at Julsundet, Halsafjorden, Trælbodneset, Gjeveneset and
Rjåneset. The strongest winds are also typically associated with southerly flow. While northeasterly winds are frequent at
Ona, the local terrain forcing at many of the observation sites typically stagnates such larger-scale flow, or rotates it along
the main fjord axes. Furthermore, northeasterly flow at Ona is presumably a result of large scale synoptic flow from a wide
sector covering flow from the northwest to the northeast, and will hence be associated with different wind directions at each
site. The sites most exposed to northeasterly flow are Åkvik, Gjeveneset and Rjåneset, while frequent and strong northerly
flow is in fact found at most of the sites, e.g. in Julsundet. In order to facilitate a more direct comparison of the wind
conditions at the sites and the variation within the region, wind roses from Ona and the sites, based on data for 1 year, are
shown in Fig. 12. Only concurrent data is used for the roses in individual panels, i.e. short periods of downtime are removed

for all sites in the same fjord. The wind roses are overlaid on the topography and highlight in a qualitatively manner the strong topographic forcing at low-levels in the fjords, as well as the large regional variations in the wind conditions for the given year. The similarity of the wind roses for Ona in Figs. 9, 11 and 12 implies that the same spatial variations exist in the regional long term wind conditions, as for the 1 year period used in Fig. 12.

Vertical wind shear may be extracted from concurrent measurements at several heights. An example from the four masts in Sulafjorden over the 1-year period (1 March 2018 – 28 February 2019) is shown in Fig. 13 (left). To the right are the mean profiles for wind speed above 10 m/s on the top sensors showing increased vertical shear. A meaningful interpretation of mean flow quantitites, such as the vertical wind shear in relation to the logarithmic wind profile, requires a more careful analysis of the observational data to be used than typically required away from complex topography. That is, the structure of the mean flow will be fundamentally different depending upon e.g. the general flow direction and speed, weather type and synoptic situation.

The monthly temperature, observed at the top most sensor in the Kvitneset mast is shown in Fig. 14, in addition to temperature observations from the Ona II reference station. There are on average small differences between the monthly temperature at both sites, with most notable difference being that the maximum temperature is typically 1-3°C higher at Kvitneset than at Ona. The observed mean monthly temperatures are also quite similar to the mean from the 18 year period. The most notable differences are that April, July and November 2018, as well as 2019 were 1-2°C warmer than average, while March 2019 was ~2°C colder. To illustrate some of the details in the data, the temperature and wind at Kvitneset during the early part of a varm day on 28 July 2018 are shown in Fig. 15. At this time there was a high pressure over the Kola peninsula and a low pressure system over the british isles, giving rise to the advection of warm air from the east which was ~20°C at 850 hPa (not shown). Skies were clear and there was presumably a large scale subsidence in the lee of the mountains of west and mid Norway. The wind was southerly and weakening during the early hours of 28 July 2018, and the lowest temperature was measured at 9 m a.g.l and the highest temperature at the top of the mast. This is indicative of a very stable boundary layer, which is cooled from below by radiatiative cooling as well as the sensible heat flux between the ocean surface and the surface layer. There are large oscillations in the temperature at upper levels, especially between 6 UTC and 8:20. These are presumably associated with the advection of warm air, which is detached from the colder air below. The top sensors are within this warm layer for long periods while the depth of the layer varies such that the sensors at 44 m and 71 m are only located inside this layer for short periods of time. The wind speed starts to increase and the vertical mixing increases between 7 and 8 UTC, and at 8:30 the colder surface air appears to be mixed up to at least 100 m but the layer is however still stably stratified. Weak winds and a varying wind direction are associated with the period of strongest solar heating from 9 until the early afternoon. There is a gradual warming of the whole layer until 12 (noon) at which time the whole layer is well mixed or only weakly stably stratified, and the wind speed has increased at many of the masts. Large variations in the vertical velocity at the top sensor appear to be associated with periods of increased mechanical and convective mixing, in particular between 10 and 11 UTC. This weak-wind case illustrates the complex interactions of air masses and topography on different scales with strong temporal and spatial variations, both in the vertical as well as the horizontal. Situations of strong wind are likewise characterized by a complex structure of the flow field, related to the proximity to the complex terrain (not shown).

Masts on both sides of the fjords allow for investigation of the simultaneous differences in the wind field on each side of the fjord. An example is given for Halsafjorden (Fig. 16). The mean wind speed is stronger at Åkvik than at Halsaneset for all wind directions except for winds from the south. The strongest winds observed at the masts are observed at Halsaneset during southerly winds, while winds are strongest at Åkvik during northwesterly flow. This is a result of the orographic forcing as well as the orientation of the fjord main axis. The mountain south of the Åkvik mast presumably introduces som sheltering while northwesterly flow may be accelerated somewhat along the terrain on the eastern side of the fjord.

As the full 3-dimensional wind vector is observed with a temporal frequency of 10 Hz, the turbulence spectral density can be estimated. An arbitrary example of such an estimate is given in Fig. 17, based on observations of a northerly storm at 50 m in the Julbø mast. The analysis is based on observations from a 20-minute period starting at 13:40 UTC on 1 January 2019. The horizontal wind vector is decomposed in components oriented along the mean wind direction, as well as perpendicular

to it. The wind speed data are linearly detrended to ensure the stationarity of the wind data and smoothed to reduce effects
from the sharp interval boundary. The spectral density is calculated using a fast Fourier transformation, implemented in a
periodogram-method in a standard signal processing package (scipy, 2020) in the python programming language. The blue
dots are the spectral energy density at individual frequencies while a 100 point running mean provides a smoother
represantation of the results. The reduction in energy density with higher frequency has a similar slope as the -5/3 power law
for turbulence spectra, i.e. as indicated by the theoretical predicion of Kolmogorov (1941). This is as expected and typical
for turbulent flow at the site.
The meteorological station Ålesund (Nørve, no. 60945) has been operational since 2009 and is used as a reference for
precipitation. Brandal station located in Sulafjorden reveals much higher precipitation than what is recorded at Nørve, both
when comparing to the average conditions during the last 10 years but also within the same year (Fig. 18). This may be
related to the proximity to the steep and high mountains at Brandal, stronger forced uplift during northerly flow and more
spillover during southerly flow.
**5 Data access**
The data are available on the MET Norway API frost.met.no (precipitation measurements at Brandal II with station number
59570) and from Arctic Data Centre (ADC): "DOI: 10.21343/z9n1-qw63" (Furevik et al., 2019). They are registered as a
data collection, as it is a dynamic data set which is growing in time. The data is typically updated on a daily basis, but data
missing in the first dissemination to the server are typically available with a lag of 1 - 3 months.
The data on ADC are posted as a file for the raw data (10 Hz) and a file for the 10-minute mean wind speed, separately for
each mast and each month. Each file contains data from the different heights at the specific mast, including self-describing
metadata, such as geographical location and sensor heights. Temperature at different heights is also posted for each month
for two mast (Kvitneset and Trælbodneset, files of type temp_0p2hz). Additional meteorological data from the weather mast
at Kvitneset, i.e. tMetpack_1hz (temperature), prsMetpack_1hz (air pressure), dewpointMetpack_1hz (dew point
temperature), RHMetpack_1hz (relative humidity) are posted in files with KvitnesetKlima in the file name.
**6 Summary**
We have presented the atmospheric part of a unique, and large, atmospheric and oceanic dataset, which is presently being
built in connection with several planned fjord crossings in the Coastal Highway E39 - project of the NPRA. The atmospheric
part of this measurement programme includes wind observations in 12 tall masts in the three different fjord systems of Mid-
Norway, and it started in 2014 and is presently ongoing. The overall data return is 98.9 %. The data collection is described,
including a short summary of the geography at the sites. Examples of observed parameters are presented and put in context
with observations and long-term conditions at reference weather stations. The examples illustrate the quality of the data, but
also a strong influence of the steep terrain on the wind measurements from these land-based masts. In addition to local
design and planning of infrastructure, the data collection may be useful for investigation of boundary flow in complex
terrain, and for verification of numerical modelling. In combination with remote sensing and oceanographic data from buoys
deployed in the project, it offers a solid basis for the study of a fjord system over at least a decade. The data collection may
furthermore be useful for the industry or in other fields of research, where wind climate is of importance.
**Author contributions.** B. R. Furevik is responsible for publication of the data set and writing of the manuscript together
with H. Ágústson. H. Ágústson is responsible for the first line of quality control and a systematic analysis of the dataset, as
well as processing of files into netCDF-format and transfer to MET Norway. A. L. Borg is responsible for furher quality
control of files, aggregation into monthly files and posting to the repository. B.R. Furevik, H. Ágústson and Z. Midjiyawa
made the analyses presented in this paper. F. Nyhammer is responsible for the design, deployment and maintainance of the
masts and instrumentation. M. Gausen is in charge of the measurement campaign for the Coastal Highway E39 project in
Mid-Norway.
**Acknowledgements.** This work and the measurement campaign is financed by the Norwegian Public Roads Administration
as part of the Coastal Highway E39 project in Mid-Norway. We acknowledge the contribution of Jørn Arve Hasselø at
NPRA, who leads the fjord crossing project, together with Magne Gausen. Knut Harstveit is acknowledged for his part in the
design of the measurement campaign, and as is Nina Elisabeth Larsgård for her part in the planning of the precipitation site
at Brandal. Map layers (used in figures 1-3, 5-8 and 12) are obtained from the Norwegian Mapping Authority
(https://kartverket.no/). The Norwegian Mapping Authority's free products are licensed under Creative Commons
Attribution 4.0 International (CC BY 4.0).

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

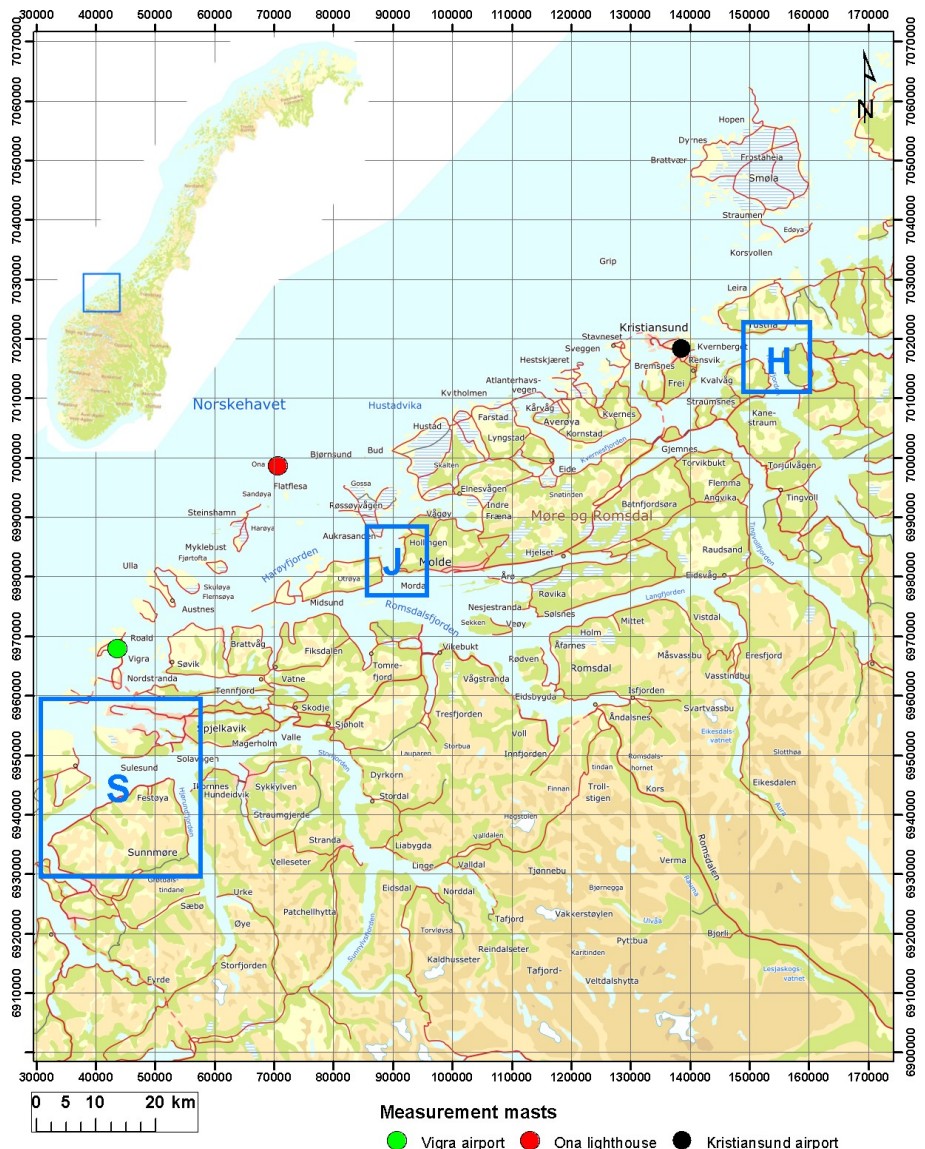

**Figure 1. Overview of a part of the Møre and Romsdal region (approximate location shown in the inset) and the**
**location of the three areas where the meteorological masts are located (S, J and H). The locations of three national**
**weather stations with long-term data available, are indicated with coloured circles. Map layers are © Kartverket and**
**licensed under Creative commons version 4.**

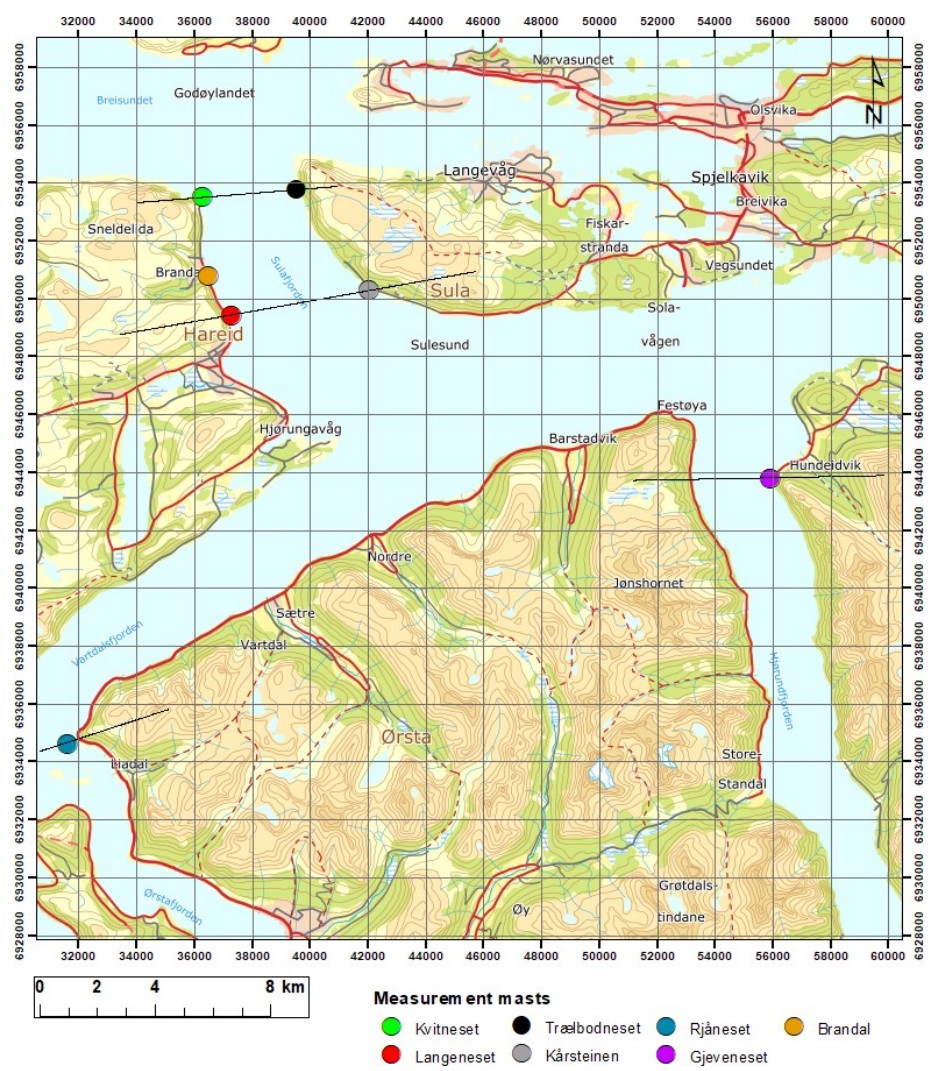

**Figure 2. Map of Storfjorden fjord system with location of the seven observational sites and height profiles shown in**
**Fig. 3. Map layers are © Kartverket and licensed under Creative commons version 4.**

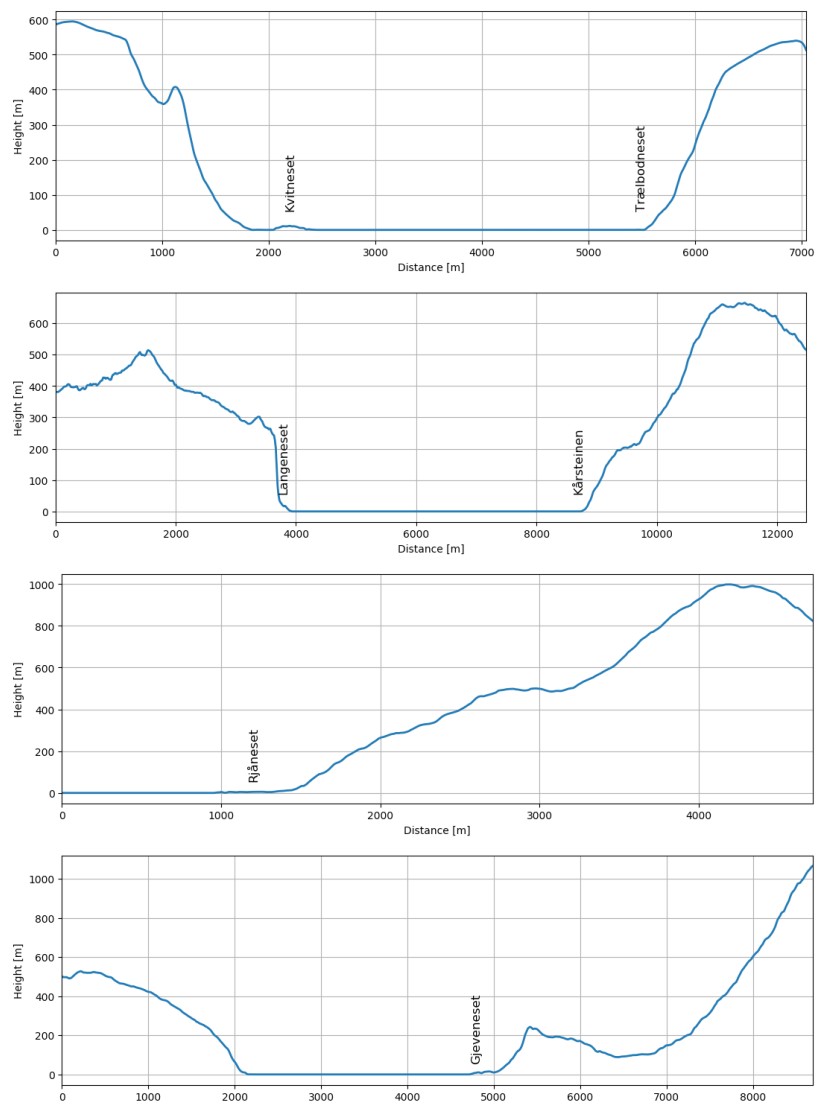

Figure 3. Terrain profiles along the sections indicated in Fig. 2, with the locations of the masts indicated. Terrain data are © Kartverket and licensed under Creative commons version 4.

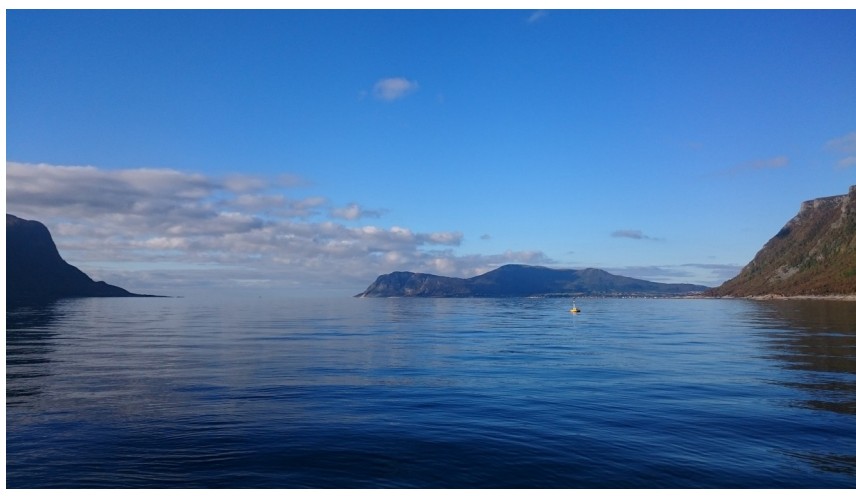

**Figure 4. Sulafjorden with the islands Hareidlandet, Godøy and Sula from left to right. Between Hareidlandet and**
**Godøy is Breisundet, which is the main opening of the fjord system to the Norwegian Sea. Photograph taken on 13**
**October 2016.**

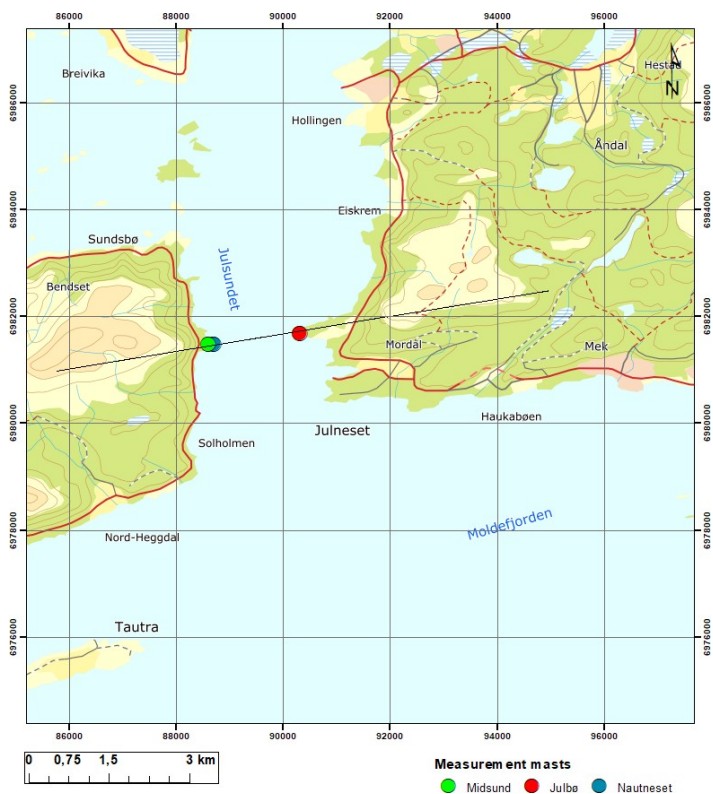

**Figure 5. Map of Julsundet with location of meteorological masts and the height profile shown in Fig. 6. Map layers**
**are © Kartverket and licensed under Creative commons version 4.**

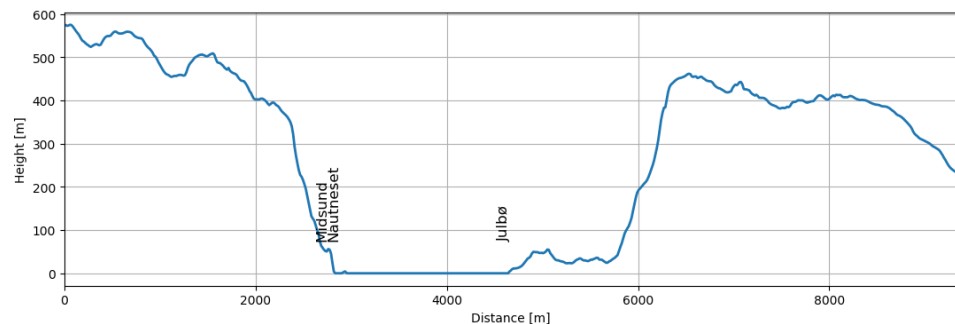

**Figure 6. Terrain profiles along the sections indicated in Fig. 5, with the locations of the masts indicated. Terrain data**
**are © Kartverket and licensed under Creative commons version 4.**

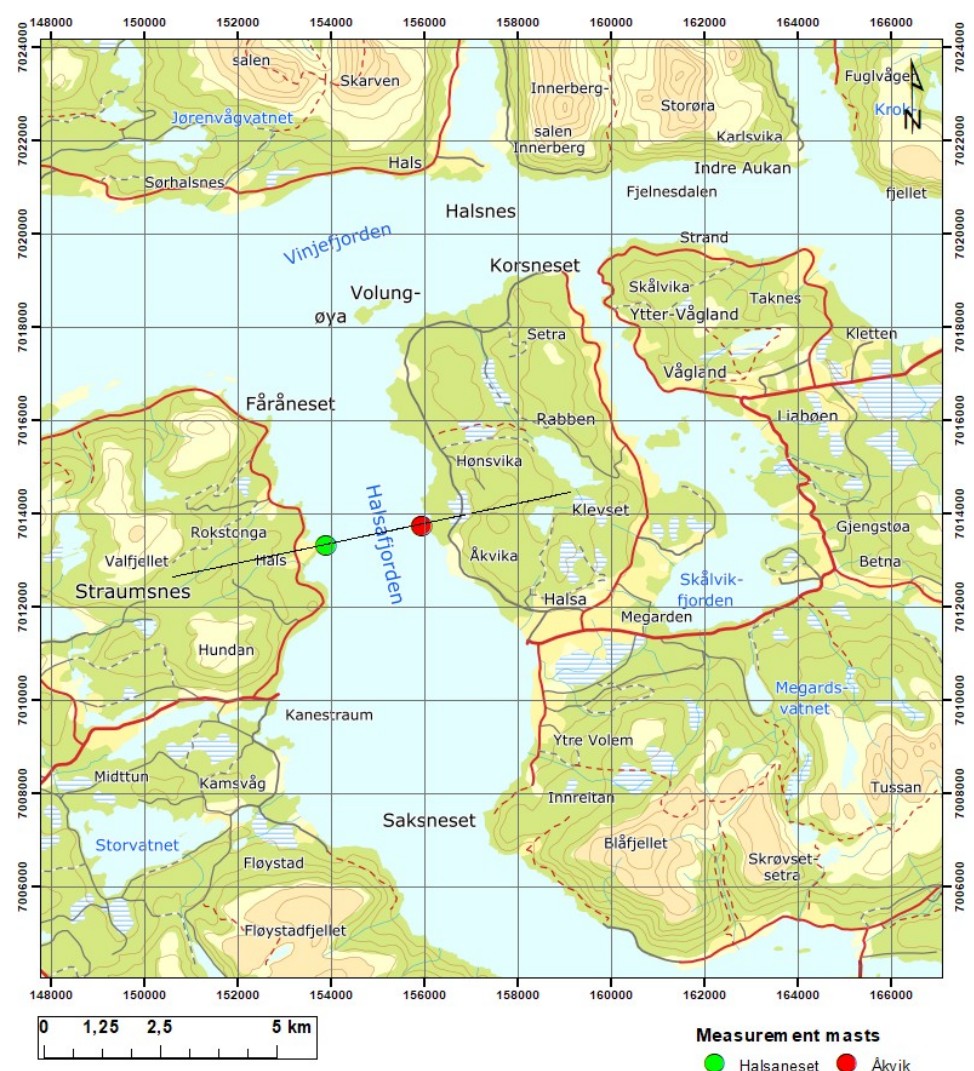

**Figure 7. Map of Halsafjorden with location of the meteorological masts, and the height profile shown in Fig. 8. The mast Åkvik2 is a contination of Åkvik and located at the exact same location. Map layers are © Kartverket and licensed under Creative commons version 4.**

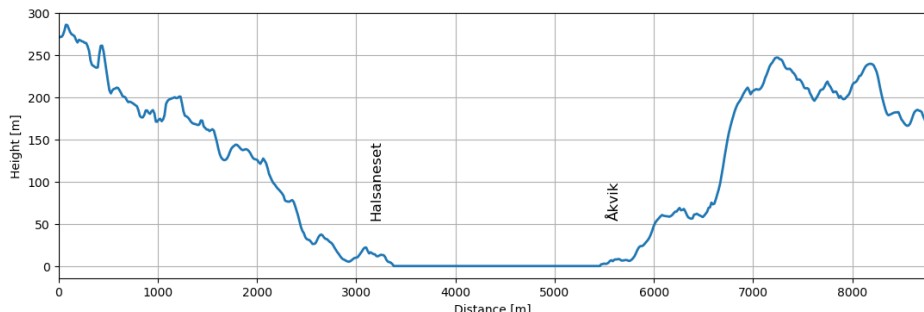

**Figure 8. Terrain profiles along the sections indicated in Fig. 7, with the locations of the masts indicated. Terrain data**
**are © Kartverket and licensed under Creative commons version 4.**

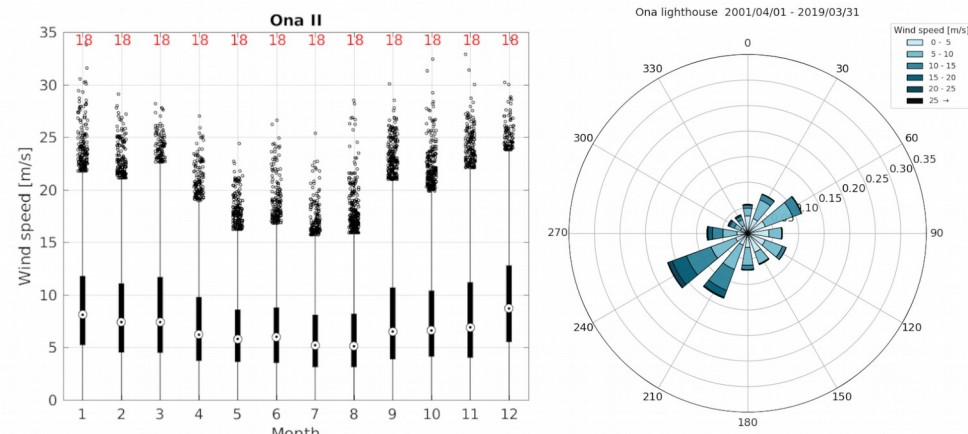

Figure 9. Wind statistics for the 18 year period at Ona II. Left: Box plot of wind speed per month of the year. The boxes in the plots shows the 25/75 percentiles with the median value as a circle inside. The lines above and below (the whiskers) represent 1.5 interquartile range from the box. Values beyond this are plotted as dots above each line. The red numbers above each month, show the number of full months used to produce each box. Right: Wind rose showing the wind speed and direction distribution. The length and direction of the bar shows the directional distribution of the wind speed while the colour scale shows the wind speed distribution.

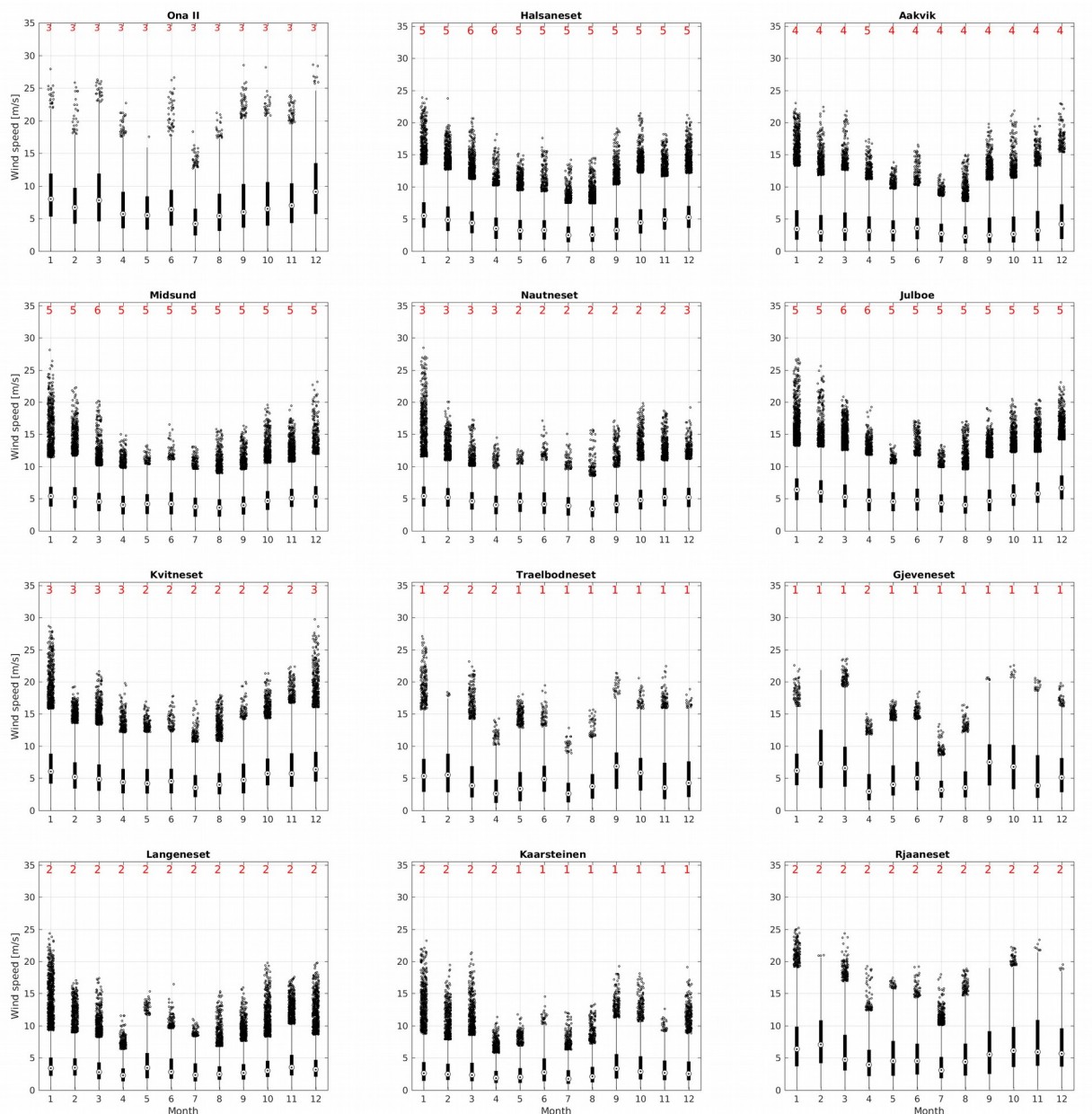

Figure 10. Box plots of wind speed per month of the year over three years from Ona II (reference station) and all available data the uppermost sensor at the sites. The time periods for each panel are found in the corresponding panel in Fig. 11.

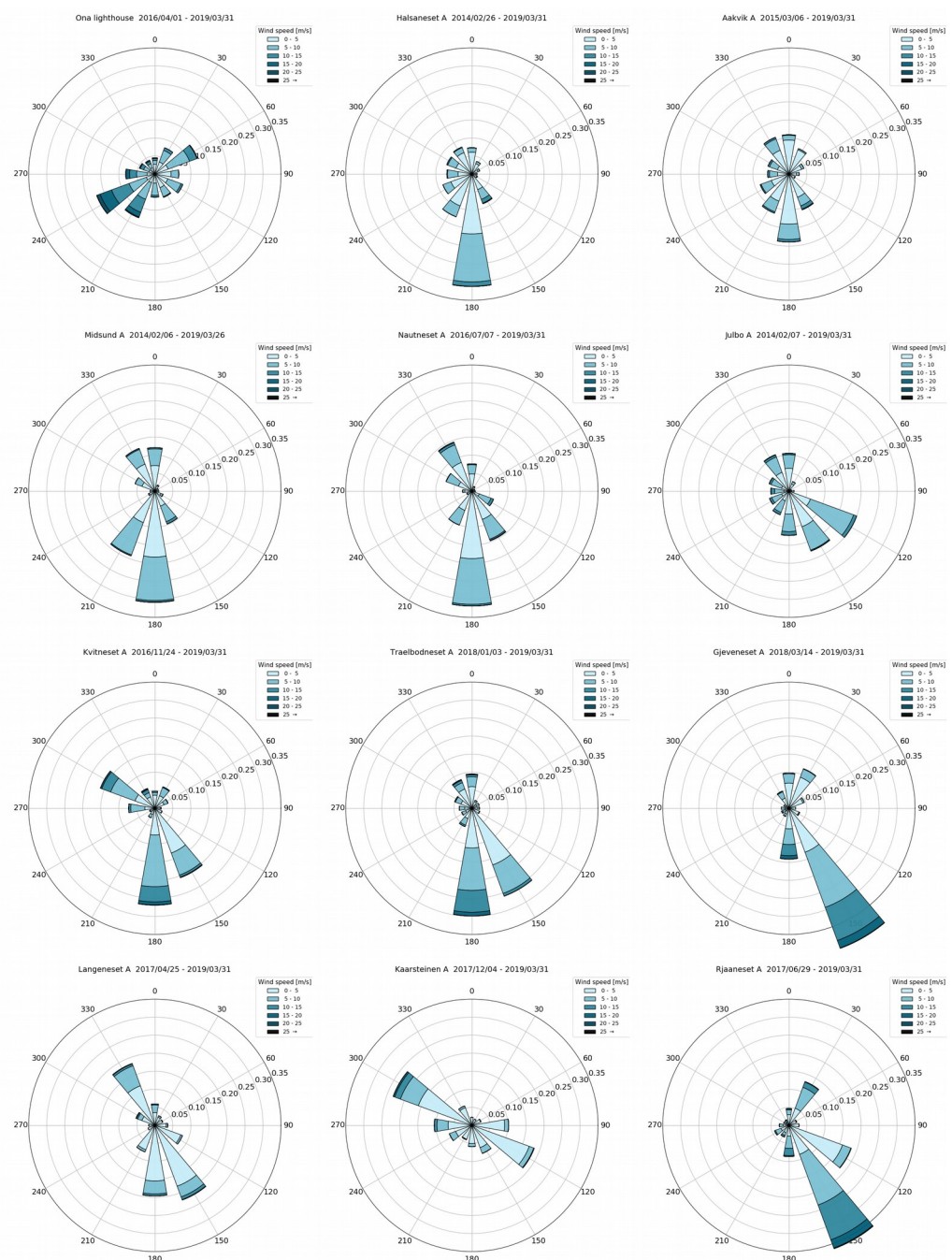

**Figure 11. Wind roses showing the wind speed and direction distribution over three years from Ona II (reference station) and all available data from the uppermost sensor at the sites.**

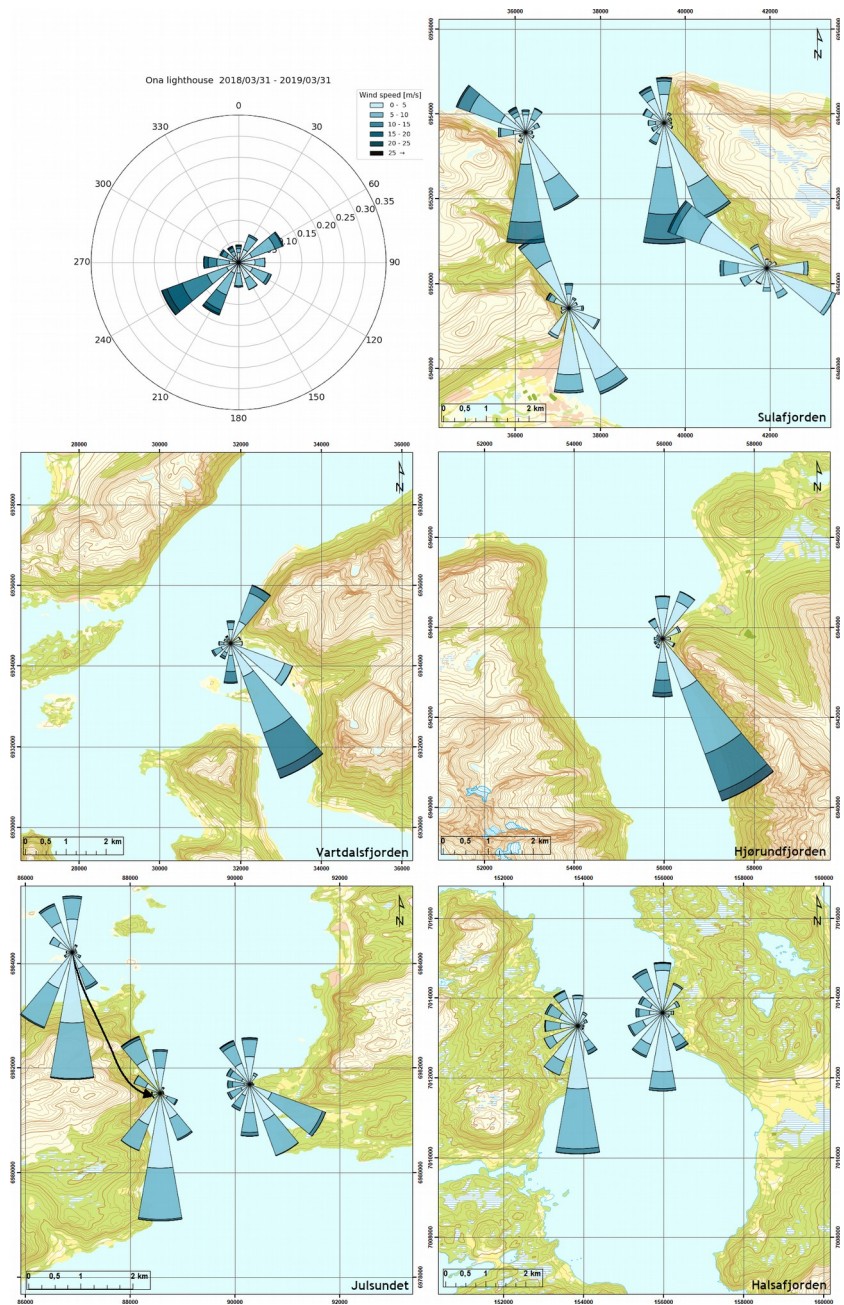

**Figure 12. Wind roses from Ona and the top-sensor of at each site, overlaid on topographic maps. Only data from**
**1 March 2018 - 28 February 2019, at all the sites, are used to produce the wind roses. All the roses are in the same**
**scale as the Ona wind rose (top left). Map layers are © Kartverket and licensed under Creative commons version 4.**

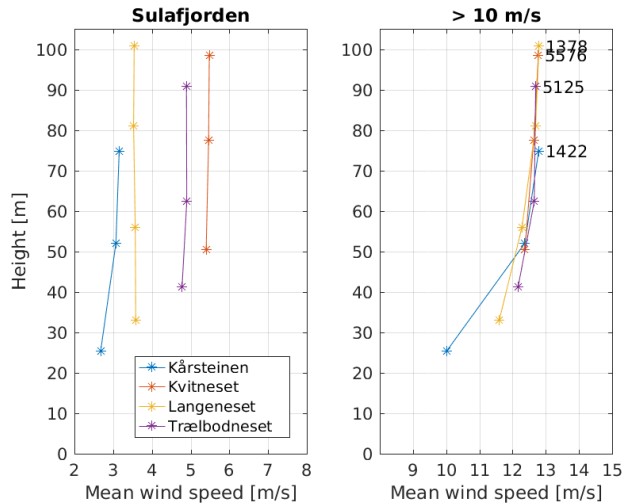

**Figure 13. Mean vertical wind shear from the four masts in Sulafjorden (left). Only data from 1 March 2018 - 28**
**February 2019 are used to produce the profiles. Height is the altitude of the sensor relative to the sea surface. Right:**
**Mean vertical wind shear for situations with wind speed higher than 10 m/s at the top sensors. The number of profiles**
**are indicated.**

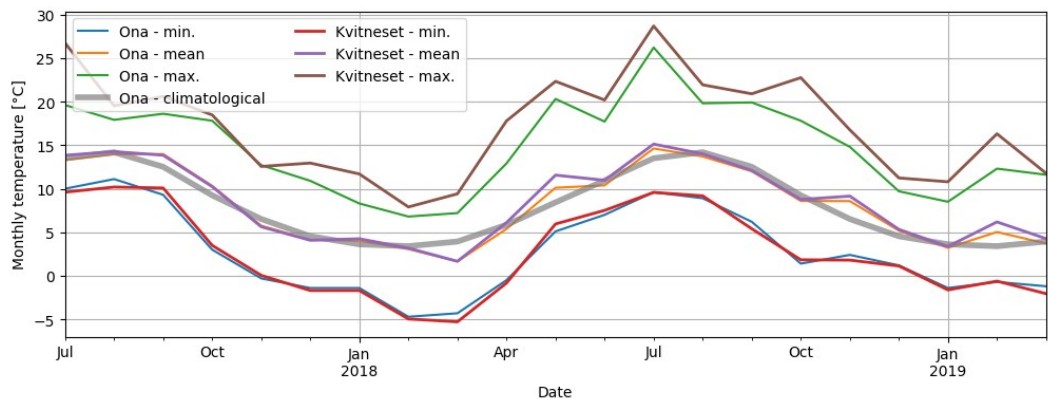

**Figure 14. Monthly mean, maximum and minimum temperature at top of Kvitneset mast and at the Ona reference**
**meteorological station. Also shown is the mean temperature (thick gray line) at Ona, for a 18 year period.**

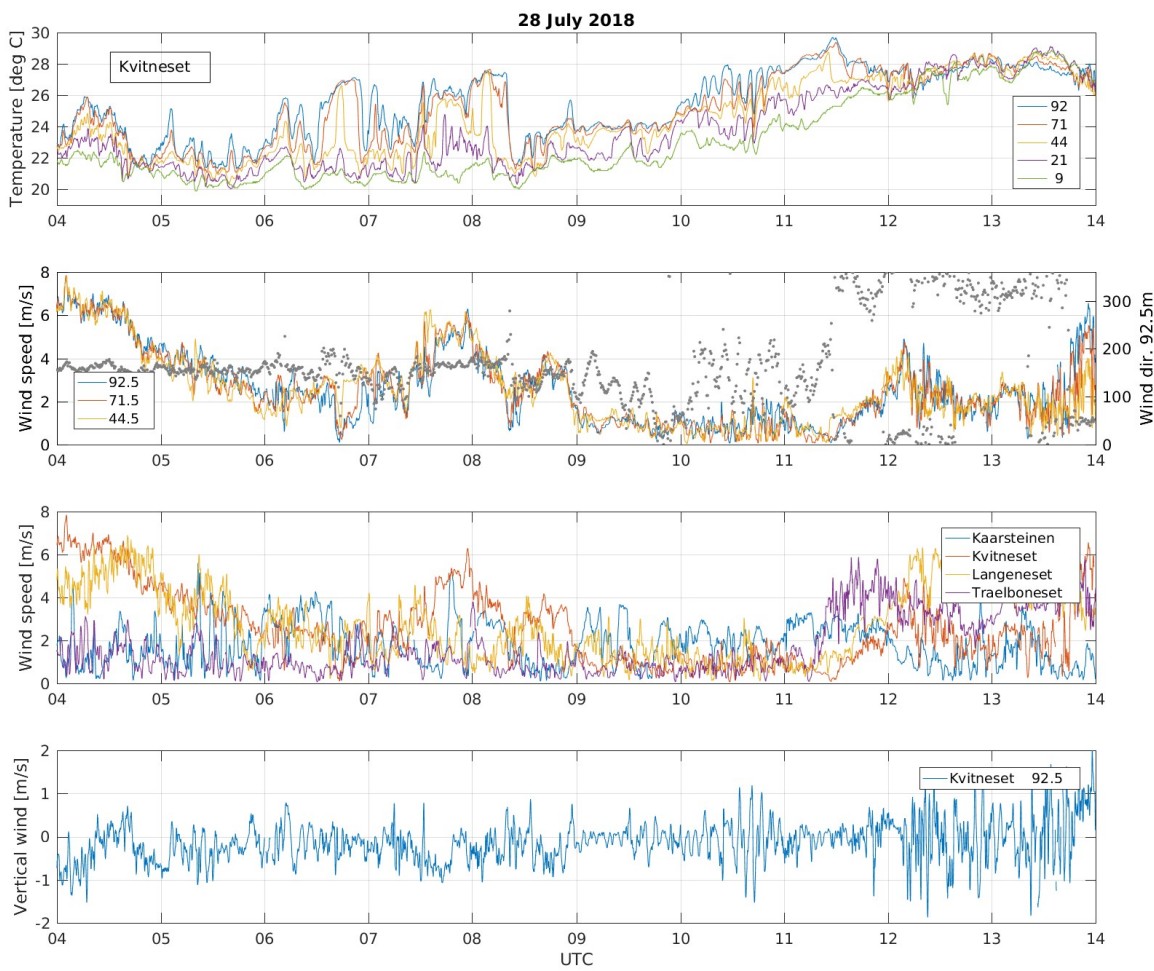

**Figure 15. Time series of temperature, wind direction, horizontal and vertical wind speed at Kvitneset (panel 1, 2 and 4 from top) and horizontal wind speed from the top sensor of all four masts in Sulafjorden (panel 3). Sensor heights at Kvitneset are given in the legends. The 10 Hz wind speed data are smoothed using a 30 s median filter.**

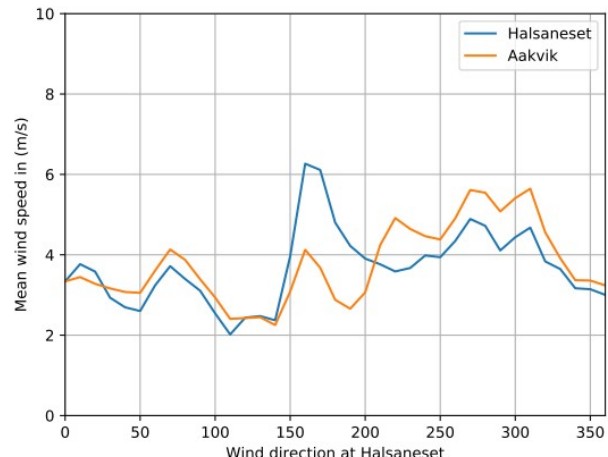

**Figure 16. Wind speed variation at Halsaneset and Åkvik in Halsafjorden, as a function of wind direction at Halsaneset on the western side of the fjord. Based on 4 years of data (2016 - 2019).**

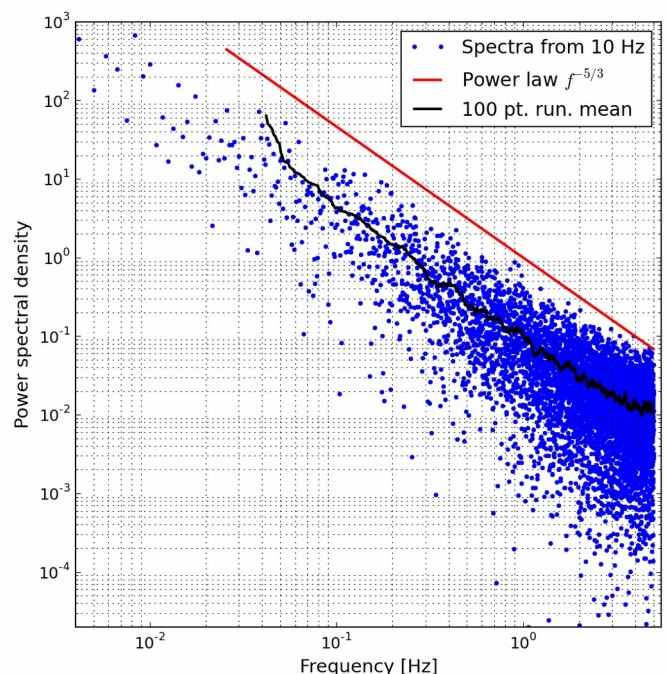

487 **Figure 17. Example of turbulence spectra for the along wind component during a northerly storm with ~25 m/s mean**
488 **wind at the top sensor of the Julbø mast. The spectra are analysed from the 20-minute period before 14:00 UTC on 1**
489 **January 2019.**

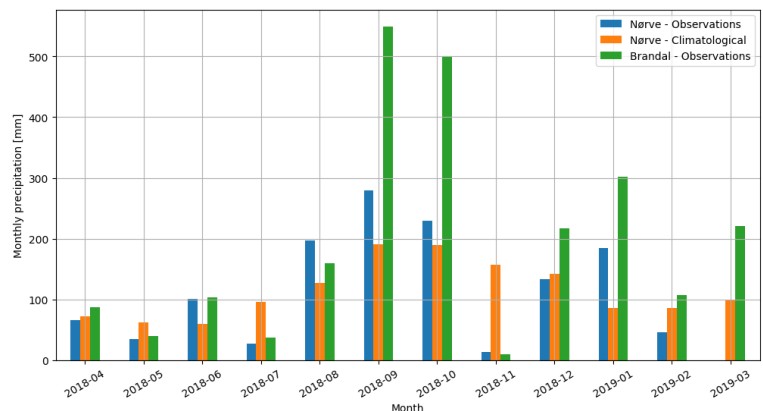

**Figure 18. Monthly measured precipitation at Brandal (green), compared to the same period (blue) and a mean for 2009 - 2019 (orange) at the reference station Nørve in Ålesund.**

Table 1: Overview of key parameters regarding the meteorological measurement sites, grouped by location. Boom direction is given as the true direction as seen from the mast, and can be used for all levels. An empty end date for the observation period implies that the observations are ongoing. Observed variables are wind speed (f) and direction (d), vertical wind speed (w), temperature (t), dew point (td), relative humidity (rh) and atmospheric pressure (prs).

| Fjord | Mast | Mast height | Ground level | Coordinates (UTM 32 / WGS84 geographical) | Observation period | Sensor heights [m] | Boom dir. | Var. |
|---|---|---|---|---|---|---|---|---|
| Sula-fjorden | Kvitneset | 100.5 m | 6 m | 6924741 N, 345142 E 62.421595° N, 6.00112° E | 2016-11-24 - | 92.5, 44.5, 71.5 | 72° | f, d, w |
| | Kvitneset temperature | | 6 m | 6924741 N, 345142 E 62.421595° N, 6.00112° E | | 21.0. 44.0, 71.0, 92.0 | | t |
| | Kvitneset Klima | | 6 m | 6924741 N, 345142 E 62.421595° N, 6.00112° E | 2017-06-27 - | 9.0 | | t, td, rh, prs |
| | Langeneset | 97.0 m | 6 m | 6920740 N, 346520 E 62.386301° N, 6.031318° E | 2017-04-26 - | 27.0, 50.0, 75.0, 94.8 | 80° | f, d, w |
| | Trælbodneset | 78.0 m | 14 m | 6925267 N, 348347 E 62.42763° N, 6.062626° E | 2018-01-03 - | 27.3, 48.3, 76.8 | 289° | f, d, w |
| | Trælbodneset temperature | 78.0 m | 14 m | 62.42763° N, 6.062626° E | | 3.0, 30.0, 50.0, 78.0 | | t |
| | Kårsteinen | 63 m | 12 m | 6922074 N, 351140 E 62.400201° N, 6.119176° E | 2017-12-04 - | 13.4, 40.0, 62.8 | 222° | f, d, w |
| | Brandal precipitation | | 27 m | 6922033 N, 345589 E | 2018-03-15 - | 1.5 | | r |
| Hjørund-fjorden | Gjeveneset | 30 m | 3 m | 6916898 N, 365563 E 62.359209° N, 6.402158° E | 2018-03-14 - | 18.5, 29.0 | 267° | f, d, w |
| Vartdals-fjorden | Rjåneset | 72.0 m | 8 m | 6905511N, 342274 E 62.248022° N, 5.963142° E | 2017-04-28 - | 28.8, 51.4, 71.5 | 278° | f, d, w |
| Julsundet | Midsund | 50m | 24m | 6957381 N, 394530 E 62.731663° N, 6.936432° E | 2014-02-11 - 2019-03-26 | 31.9, 12.7, 50.3 | 73° | f, d, w |
| | Julbø | 50 m | 4 m | 6957730 N, 396210 E 62.735273° N, 6.969062° E | 2014-02-14 - | 12.7, 31.9, 50.3 | 233° | f, d, w |
| | Nautneset | 68 m | 2 m | 6957381 N, 394634 E 62.731693° N, 6.938466° E | 2016-11-10 - | 32.7, 52.3, 68.3 | 238° | f, d, w |
| Halsa-fjorden | Halsaneset | 50 m | 4 m | 6995095 N, 456472 E 63.082697° N, 8.138198° E | 2014-02-26 - | 12.7, 31.9, 50.3 | 104° | f, d, w |

| | Åkvik | 50 m | 6 m | 6995697 N, 458519 E 63.08834° N, 8.178568° E | 2015-03-06 2020-05-08 | 17.0, 31.9, 48.3 | 225° | f, d, w |
| | Åkvik2 | 100 m | 6 m | 6995697 N, 458519 E 63.08834° N, 8.178568° E | 2020-05-09 - | 48.3, 78.1, 97.2 | 225° | f, d, w |

**Table 2: Main statistics of wind data set at top sensor, including mean, median, maximum wind speed and 99th percentile of wind speed, the maximum gust (3 s), as well as the 99th percentile of the up/down vertical wind gust [ms⁻¹].**

| Fjord | Mast | Height [m] | Mean wind speed | Median wind speed | Maximum wind speed | 99th perc. of wind speed | Max. gust | 99th perc. vert. gust |
|---|---|---|---|---|---|---|---|---|
| Sulafjorden | Kvitneset | 92.5 | 5.64 | 5.03 | 29.70 | 16.52 | 37.0 | -13.4 / 8.6 |
| | Langeneset | 94.8 | 3.59 | 2.95 | 24.34 | 13.26 | 37.3 | -13.6 / 7.4 |
| | Trælbodneset | 76.8 | 5.01 | 4.24 | 27.04 | 15.97 | 46.1 | -9.2 / 7.1 |
| | Kårsteinen | 62.8 | 3.17 | 2.39 | 23.21 | 12.97 | 32.1 | -8.6 / 6.3 |
| Hjørundfjorden | Gjeveneset | 29.0 | 5.85 | 4.83 | 23.55 | 17.82 | 43.6 | -6.3 / 5.7 |
| Vartdalsfjorden | Rjåneset | 71.5 | 6.04 | 5.04 | 25.18 | 17.34 | 41.2 | -6.8 / 6.3 |
| Julsundet | Midsund | 50.3 | 4.61 | 4.45 | 28.15 | 11.75 | 40.0 | -7.4 / 6.2 |
| | Julbø | 50.3 | 5.47 | 5.15 | 26.74 | 14.14 | 39.6 | -4.8 / 5.0 |
| | Nautneset | 68.3 | 4.80 | 4.59 | 28.46 | 12.83 | 41.9 | -9.0 / 6.1 |
| Halsafjorden | Halsaneset | 50.3 | 4.30 | 3.91 | 23.87 | 12.62 | 35.1 | -5.0 / 4.3 |
| | Åkvik | 48.3 | 3.80 | 3.03 | 23.00 | 12.94 | 34.4 | -3.5 / 4.8 |