# Peer review of "Meteorological observations in tall masts for mapping of atmospheric flow in Norwegian fjords"

_Earth System Science Data, 2020_

## Short Comment (SC1) · 6 Jul 2020

The manuscript "Meteorological observations in tall masts for mapping of atmospheric flow in Norwegian fjords" by Furevik et al. deals with a unique data set, freely accessible since 2018, which is particularly valuable for both engineers and scientists working on the E39 Coastal Highway Route (Ferry Free E39). Nevertheless, some of the statements in the manuscript may be unclear, ambiguous or misleading:

- Line 59, the authors mention that the "dataset provides invaluable data describing the atmospheric forcing, both climatic and short-term, pertaining to the technical

design of large structures in complex terrain." Although I understand the enthusiasm of the authors, one should keep in mind that the potential and limits of the dataset have not yet been assessed in details. It is also unclear to me what the authors mean exactly by "atmospheric forcing, both climatic and short-term" with respect to structural design. A more specific reformulation would be welcome.

- Line 34-35. As the authors already know, there has been a similar campaign in the Bjørnafjord since 2015. Although the data in that fjord are not publicly available, it may be useful to the reader to know that the campaigns in the Sulafjord, Halsafjord and Julsundet are not the only ones.

- Line 98: If no filtering is applied beforehand, downsampling a time series from 20 Hz to 10 Hz will amplify aliasing not reduce it. In general, downsampling increases aliasing. As far as I know, the downsampling procedure was done without filtering, resulting in undesirable aliasing, visible in Figure 15, at frequencies above 4 Hz.

- It may be informative to the reader to know if the high-frequency sonic temperature is freely available or not. I am aware that some 2-Hz sonic temperature records are usable, but this sampling frequency may be too low to study turbulent fluxes. A sampling frequency of 10 Hz or more is desirable for such purposes.

---

## Author Comment (AC1) · 26 Jul 2020

Thank you for your interest and the comments provided. We have copied the four comments here and reply to them one by one. The information will also be included in a revised manuscript.

Comment 1: Line 59, the authors mention that the "dataset provides invaluable data describing the atmospheric forcing, both climatic and short-term, pertaining to the design of large structures in complex terrain." Although I understand the enthusiasm of the authors, one should keep in mind that the potential and limits of the dataset have not yet been assessed in details. It is also unclear to me what the authors mean exactly

by "atmospheric forcing, both climatic and short-term" with respect to structural design. A more specific reformulation would be welcome.

Reply: We agree that the sentence can be rephrased and we will certainly consider this in the revised version of the manuscript. The message we sought to convey was that the dataset is unique in both the length as well as in the detail of the observed time series at the available sites. The series are long enough so that they can be of use in describing the climatic conditions at the sites, but they are also detailed enough to describe well single weather events of interest and capture some of the complexity in the flow structure on either side of the planned crossings. We do not insist that the mast data can be easily extrapolated to describe the conditions at the middle of the fjord crossings. In fact, the data set of wind, temperature, precipitation etc. presented in this manuscript is an essential part of a larger data set, including Wind Lidars and buoys, which is created in order to collect as much meteorological and oceanographic information from the whole crossing area as is technically and economically feasible. The planned crossings are longer and in more difficult terrain than previously built crossings, and they may require design and solutions not used before. At the start of the project in 2014, it was not clear what atmospheric and oceanic data would be needed and what aspects of the climate/weather had to be accounted for. A lot has been learned since then, but it is correct that we do not know yet how valuable the mast data will be in itself for the design. It is possible that parts of the detailed design calculations may have to be based on other data. However, the wind measurements have proved invaluable in several other aspects, for example to verify numerical models (see e.g. Midtbø et al. (2020) MET report 05-2020 "Finskala modellering av vind i fjorder. Sulafjorden og Vartdalsfjorden 2018" available at https://tinyurl.com/y4m7xqd8 ). Using numerical models which represent the local wind conditions, we can relate it to the large scale flow (historical, i.e. climatic) which provides wind statistics for design, and this will also help to improve forecasting during a construction phase.

Comment 2: Line 34-35. As the authors already know, there has been a similar campaign in the Bjørnafjord since 2015. Although the data in that fjord are not publicly available, it may be useful to the reader to know that the campaigns in the Sulafjord, Halsafjord and Julsundet are not the only ones.

Reply: We agree that we should include the information here that extensive measurement campaigns related to E39 also exist in other fjords even if those data are not freely available.

Comment 3: Line 98: If no filtering is applied beforehand, downsampling a time series from 20 Hz to 10 Hz will amplify aliasing not reduce it. In general, downsampling increases aliasing. As far as I know, the downsampling procedure was done without filtering, resulting in undesirable aliasing, visible in Figure 15, at frequencies above 4 Hz.

Reply: When the output of the Gill Windmaster Pro is set to 10 Hz, the sampling is done at 20 Hz. Each output value is based on the average of two ultrasonic samples, and this averaging acts as a filter.

Comment 4: It may be informative to the reader to know if the high-frequency sonic temperature is freely available or not. I am aware that some 2-Hz sonic temperature records are usable, but this sampling frequency may be too low to study turbulent fluxes. A sampling frequency of 10 Hz or more is desirable for such purposes.

Reply: The 10 Hz temperature measurement from some of the sonic anemometers was stored. However, it is not a part of the available dataset on thredds and is hence not discussed in the manuscript.

---

## Referee Comment (RC1) · Anonymous Referee #1 · 27 Jul 2020

Overall, this is an excellent paper. The data are easily accessible and well-organized. The dataset is well-described and the justification is clear for collecting such data. Revisions are very minor, and the paper will be an excellent introduction to the dataset.

Line 242: First sentence here is ambiguous. Does this mean that December wind speeds are 8-9 ms-1, or is that the annual median, with higher speeds in winter?

Line 280: It is not clearly explained why this turbulence data is being presented or how to interpret it. Some language could be added here to explain how this information can be useful to a user. Is this particular sample of data being shown because it is particularly interesting, or just an example of the larger set?

---

## Referee Comment (RC2) · Anonymous Referee #2 · 8 Aug 2020

This paper presents a dataset of meteorological observations collected in 11 tall masts in three different fjords systems of Mid-Norway. A large part of the manuscript is devoted to the description of each measurements site, which include useful information about fjords geographic features and operating instruments. The last two sections are dedicated to a (too) brief description of quality control procedures and to a presentation of measured wind, temperature and precipitation data.

As general comment, I think that the authors present an interesting dataset, which can be certainly useful for meteorological and engineering purposes. However, I think that this work has some point of weaknesses that must be addressed before considering

it for publication in ESSD. First, the quality presentation of the study is unsatisfactory for the level of a journal such as ESSD: therefore, the first suggestion is to perform a formal revision of the manuscript, improving language and style.

From a strictly scientific point of view, the paper must be revised according to the following suggestions:

- In my opinion, when presenting the data (Section 4) authors use the word "climate" in an inappropriate way. For example, a period of 18 years (Lines 360) cannot be used to reach any robust conclusions from a strictly climatological perspective. You can speak about climate only when you managed a meteorological time series of at least 30 year. This consideration is obviously and even more so valid for the new dataset presented in this study. For example, at Line 258 you cannot speak about "wind climate", considering only two or three years of data. I suggest to use "wind regime" and to underline that no climatological results or conclusions can be achieved from the available data. You can present your results only from a meteorological perspective. In other words, the wind regime observed in the 11 sites might be affected by the atmospheric variability and anomalies observed in a specific year and/or season, due to the very limited time period taken into account.

- In section 3, the authors describe the data handling and quality. I suggest extending this section, providing more details about data quality control, which is a critical and focal point of any data description paper. My recommendation is to structure the quality control into at least three different step, considering the following tests: 1. Gross error test, which flag data that are above or below acceptable physical limits; 2. The tolerance test, which detects the outliers, i.e. the values that are above or below some specific limits defined according to a probability distribution model; 3. The temporal coherence test, which identifies unrealistic "jumps" between two consecutive observations according to the change that might be expected for a determined variable in a specific time interval. A graphical example for each of the just mentioned basic quality control step should be provided. Moreover, the authors may also consider to apply a

fourth quality control step, based on spatial consistency among the available measurements. A useful reference may be following paper, recently published on ESSD:

Capozzi, V., Cotroneo, Y., Castagno, P., De Vivo, C., and Budillon, G.: Rescue and quality control of sub-daily meteorological data collected at Montevergine Observatory (Southern Apennines), 1884–1963, Earth Syst. Sci. Data, 12, 1467–1487, https://doi.org/10.5194/essd-12-1467-2020, 2020.

Other useful references:

Hubbard, K., You, J., and Shulski, M.: Toward a Better Quality Control of Weather Data, Practical Concepts of Quality Control, edited by: Saber, M. and Nezhad, F., ISBN: 978-953-51-0887-0, InTech, https://doi.org/10.5772/51632, 2012. 

Steinacker, R., Mayer, D., and Steiner, A.: Data Quality Control Based on Self-Consistency, Mon. Weather Rev., 139, 3974–3991, https://doi.org/10.1175/MWR-D-10-05024.1, 2011. 

World Meteorological Organization: Guide to Meteorological Instruments and Methods of Observation, 2008 Edition, WMO-no. 8 (Seventh edition), available at: https://www.wmo.int/pages/prog/www/IMOP/publications/CIMO-Guide/OLD-pages/CIMO_Guide-7th_Edition-2008.html (last access: 1 October 2019), 2008. 

- About the comparison between reference station and data from masts, I suggest to produce plots based on the same period. I understand that the data availability may be a problem, because it varies from a measurements point to another, but it is necessary to identify a common period allowing performing a real comparison between the wind roses presented in Fig. 11. When discussing this figure, I think that is important to highlight that northeastern winds have a relevant frequency only in Aakvivk A, Gjeveneset A and Rjaaneset A. How do you explain this result? Why in other mast locations the wind regime is so different from the reference one (upper left panel of Fig. 11)?

- In the introduction section (Lines 48-49), the authors claim that the measurement

campaign presented in their work may have interesting and relevant implications for studies concerning the boundary layer variability in complex terrain. I agree with the authors, but I do not understand why the authors did not further stress this point when presenting the data in section 4. Therefore, I suggest showing some examples of vertical profiles of wind speed, wind direction, temperature and relative humidity obtained from the available measurements. For example, the authors may produce a vertical profile for each of three fjords, considering the measurements that are best suited for this purpose. To highlight the good potential of the dataset, the authors may also present, only for illustrative purposes, a comparison between vertical profiles obtained in different meteorological scenarios.

Best regards.

―――――――――――――――――――

---

## Referee Comment (RC3) · Anonymous Referee #3 · 15 Aug 2020

This manuscript presents details of the meteorological observation system at the Norwegian fjords. As a part of coastal highway project, wind is the focused parameter measured at multiple vertical levels at 11 locations extending over 2-10 years of record. The dataset consists of high frequency and long-term measurements of wind speed and direction as well as other meteorological fields at selected locations. I think that the dataset is of significance to the research and engineering communities, and the manuscript covers basics of a data description paper but I have a few suggestions to improve.

1. Figure 11: I recommend plotting the wind rose at 11 stations overlaid and pointing

to the location on the map to give a perspective of the entire region. Like the example figure (with partial stations) 2. Quality control is an important part of data description. I think that Section3 Data handling and Quality should be dedicated to quality control. The current content is mainly on processing and transmission, that seems to fit in Section 5. What is the latency of the data—near real time? 3. Figure 13: How long does the observation record go back at the reference site? Is it available before 2009? If so, it would be nice to use longer time mean. 4. Line 246-247: I don't understand this sentence, please clarify.
* * *
[Figure]

[Figure]

**Fig. 1.**

---

## Author Response (AR1)

*Overall, this is an excellent paper. The data are easily accessible and well-organized. The dataset is well-described and the justification is clear for collecting such data. Revisions are very minor, and the paper will be an excellent introduction to the dataset.*

**Response: We appreciate the effort of the reviewer and the positive review. We have revised the manuscript according to the comments of the reviewer, as stated below. Additionally, some more changes have been done to the manuscript, to further improve the written language and the structure of the paper.**

*Line 242: First sentence here is ambiguous. Does this mean that December wind speeds are 8-9 ms-1, or is that the annual median, with higher speeds in winter?*

**Response: We agree that the sentence was unclear.**

**Changes:  The sentence is rewritten and now reads:** "For this period, the median wind speed at Ona is 6.6 ms$^{-1}$ which varies from 5.1 ms$^{-1}$ in August up to 8.7 ms$^{-1}$ in January (Fig. 9)".

*Line 280: It is not clearly explained why this turbulence data is being presented or how to interpret it. Some language could be added here to explain how this information can be useful to a user. Is this particular sample of data being shown because it is particularly interesting, or just an example of the larger set?*

**Response: We agree with the reviewer that the text was unclear, and as was the motive for showing the figure.**

**Changes: We have rephased the paragraph (paragraph 7 in section 4) to provide some more details and explanation.  The actual data was arbitrarily chosen to show an example of the use of the data, and that is now clearly stated in the text.**

*This paper presents a dataset of meteorological observations collected in 11 tall masts in three different fjords systems of Mid-Norway. A large part of the manuscript is devoted to the description of each measurements site, which include useful information about fjords geographic features and operating instruments. The last two sections are dedicated to a (too) brief description of quality control procedures and to a presentation of measured wind, temperature and precipitation data.*

*As general comment, I think that the authors present an interesting dataset, which can be certainly useful for meteorological and engineering purposes. However, I think that this work has some point of weaknesses that must be addressed before considering it for publication in ESSD. First, the quality presentation of the study is unsatisfactory for the level of a journal such as ESSD: therefore, the first suggestion is to perform a formal revision of the manuscript, improving language and style.*

**Response: We appreciate the effort of the reviewer and the constructive criticism. We have revised the manuscript according to all received comments and worked on the structure and the language. We believe that the manuscript has improved significantly. Response to each comment from the reviewer and a description of related changes are given below.**

*From a strictly scientific point of view, the paper must be revised according to the following suggestions:*

*In my opinion, when presenting the data (Section 4) authors use the word "climate" in an inappropriate way. For example, a period of 18 years (Lines 360) cannot be used to reach any robust conclusions from a strictly climatological perspective. You can speak about climate only when you managed a meteorological time series of at least 30 year. This consideration is obviously and even more so valid for the new dataset presented in this study. For example, at Line 258 you cannot speak about "wind climate", considering only two or three years of data. I suggest to use "wind regime" and to underline that no climatological results or conclusions can be achieved from the available data. You can present your results only from a meteorological perspective. In other words, the wind regime observed in the 11 sites might be affected by the atmospheric variability and anomalies observed in a specific year and/or season, due to the very limited time period taken into account.*

**Response:** It's correct that WMO recommends using the latest 30 years where the last year ends with 0, for calculating climate normal. This is not what we intended to do, or imply we were presenting. We are merely aiming to compare the longest available time series from the closest long-term stations to our data, in order to put the measurements from the campaign in perspective with the regional long-term wind conditions. This highlights the impact of the topography on the local wind conditions, and reveals that the "wind regime" during the period of the observations of the masts is not so different from the climate estimate based on 18 years. We certainly agree that it is wrong to use the word "climate" for a period of only a few years. However, we will insist that 15-20 years of recent observations of wind can represent well the current wind climate at a given site, but such a climate estimate should certainly not be compared to the previously mentioned climate normals typically calculated.

**Changes:** We have included a new figure (Fig. 12) and changed accordingly the formulation in the first paragraph of section 4. The discussion of climate in other parts of the document have also been changed accordingly, e.g. the word climate is not used when investigating the mean wind conditions based on the short observation series from the sites.

*In section 3, the authors describe the data handling and quality. I suggest extending this section, providing more details about data quality control, which is a critical and focal point of any data description paper. My recommendation is to structure the quality control into at least three different step, considering the following tests: 1. Gross error test, which flag data that are above or below acceptable physical limits; 2. The tolerance test, which detects the outliers, i.e. the values that are above or below some specific limits defined according to a probability distribution model; 3. The temporal coherence test, which identifies unrealistic "jumps" between two consecutive observations according to the change that might be expected for a determined variable in a specific time interval. A graphical example for each of the just mentioned basic quality control step should be provided. Moreover, the authors may also consider to apply a fourth quality control step, based on spatial consistency among the available measurements. A useful reference may be following paper, recently published on ESSD:*
*Capozzi, V., Cotroneo, Y., Castagno, P., De Vivo, C., and Budillon, G.: Rescue and quality control of sub-daily meteorological data collected at Montevergine Observatory (Southern Apennines), 1884–1963, Earth Syst. Sci. Data, 12, 1467–1487, https://doi.org/10.5194/essd-12-1467-2020, 2020.*

*Other useful references:*

*Hubbard, K., You, J., and Shulski, M.: Toward a Better Quality Control of Weather Data, Practical Concepts of Quality Control, edited by: Saber, M. and Nezhad, F., ISBN: 978-953-51-0887-0, InTech, https://doi.org/10.5772/51632, 2012. *

*Steinacker, R., Mayer, D., and Steiner, A.: Data Quality Control Based on Self-Consistency, Mon. Weather Rev., 139, 3974–3991, https://doi.org/10.1175/MWR-D-10-05024.1, 2011. *

*World Meteorological Organization: Guide to Meteorological Instruments and Methods of Observation, 2008 Edition, WMO-no.*
*8 (Seventh edition), available at:*
*https://www.wmo.int/pages/prog/www/IMOP/publications/CIMO-Guide/OLD-pages/CIMO_Guide-7th_Edition-2008.html (last access: 1 October 2019), 2008. *

**Response: The data are made available on thredds at MET for potential users to use as fit. No additional filtering of the dataset can be done by the authors in connection with producing this manuscript, and the public access to the dataset does not depend on the current manuscript. That is, the data is provided as is, and more filtering is beyond the scope of this paper and the budget for making the data available. We believe that this does not render the dataset less valuable for the potential user, and that this description of the dataset should be published in order to facilitate the use of the dataset. However, the end-user must assess his need for further quality control and put effort into the additional filtering routines necessary for his intended use of the dataset.**

**The filtering already done is described and for more in-depth filtering routines, the reader is directed to a suggestion of papers. The raw observational data for variables other than wind are made available as is, with only a first screening performed. No 10-minute values are made available for these variables. This is now stated in the section. For the 10 Hz observations of wind speed and direction, the filtering is as follows and described in the manuscript: Unphysical values beyond the specifications of the instruments are removed. Noise and spikes, i.e. unphysical jumps in the 10 Hz data, are filtered and removed. Locked values, i.e. repeated and constant values are identified and removed, but such values are somewhat frequent from the instruments. This filtering captures a large part of three of the suggested tests. No spatial testing is done and the authors are not familiar with how well it performs in complex terrain where the flow at one site may often to a large degree be "detached" from the state of the flow at other masts in the region, or even at a nearby mast in the same fjord.**

**Changes: The whole section has been improved and more details are given in the revised manuscript. Some parts were moved to section 5, i.e. description on access to the data.**

*About the comparison between reference station and data from masts, I suggest to produce plots based on the same period. I understand that the data availability may be a problem, because it varies from a measurements point to another, but it is necessary to identify a common period allowing performing a real comparison between the wind roses presented in Fig. 11. When discussing this figure, I think that is important to highlight that northeastern winds have a relevant frequency only in Aakvivk A, Gjeveneset A and Rjaaneset A. How do you explain this result? Why in other mast locations the wind regime is so different from the reference one (upper left panel of Fig. 11)?*

**Response:** We agree with the reviewer that it is problematic to compare observations from different periods. We have changed the text as described below, where we highlight that the northeastern flow at Ona is in fact a collection of synoptic scale flow from a wide sector, including flow from the northwest to northeast. That is, orographic forcing of the large scale terrain of western Norway, results in the observed flow at coastal stations typically being along the coast. This implies that northeasterly flow at Ona can be associated with a lot of different directions at the different stations, as now mentioned in the paper. An investigation of the coupling of the wind direction between the reference station and individual stations is beyond the scope of this paper.

**Changes:** We have included a new figure (Fig 12) in the updated manuscript, but we also choose to keep the original wind roses in order to present the statistics for as long observational periods as possible at each site.

In the new figure there are separate panels for each of the fjords. Wind roses are made for each mast and the Ona reference stations, and only using concurrent data for March 2017 - 2018. The wind roses are overlaid on the terrain, hence highlighting in a qualitative manner the terrain forcing at each site. As only concurrent data is used, wind conditions at all masts can be compared to each other. The text has been changed accordingly and improved considerably (4th paragraph of section 4).

The updates to the text in paragraph 4 of section 4, include an explanation for the different directional distributions at the sites, which are first and foremost due to the orographic forcing.

*In the introduction section (Lines 48-49), the authors claim that the measurement campaign presented in their work may have interesting and relevant implications for studies concerning the boundary layer variability in complex terrain. I agree with the authors, but I do not understand why the authors did not further stress this point when presenting the data in section 4. Therefore, I suggest showing some examples of vertical profiles of wind speed, wind direction, temperature and relative humidity obtained from the available measurements. For example, the authors may produce a vertical profile for each of three fjords, considering the measurements that are best suited for this purpose. To highlight the good potential of the dataset, the authors may also present, only for illustrative purposes, a comparison between vertical profiles obtained in different meteorological scenarios.*

**Response:** We agree that the potential of the data can be better visualized. Instead of wind profiles, we have chosen to include an example of time series (Fig. 14) from some of the masts in Sulafjorden, since we find that this is a better illustration of the details that the measurements can represent.

**Changes:** A new figure (Fig 14) is included and is discussed in an updated and re-phrased paragraph (fifth) in section 4. Note that in connection with the new text, then we have moved the paragraph on precipitation to be the last in the section.
This manuscript presents details of the meteorological observation system at the Norwegian fjords. As a part of coastal highway project, wind is the focused parameter measured at multiple vertical levels at 11 locations extending over 2-10 years of record. The dataset consists of high frequency and long-term measurements of wind speed and direction as well as other meteorological fields at selected locations. I think that the dataset is of significance to the research and engineering communities, and the manuscript covers basics of a data description paper but I have a few suggestions to improve.

**Response: We appreciate the effort of the reviewer and the positive review. We have revised the manuscript according to the comments of the reviewer, and our response and change related to each item is given below.**

1. Figure 11: I recommend plotting the wind rose at 11 stations overlaid and pointing to the location on the map to give a perspective of the entire region. Like the example figure (with partial stations)

**Response: We certainly agree with the reviewer.**

**Changes: We have provided a new figure as suggested (figure 12), with separate panels for each of the fjords.  Wind roses are made for each mast and the Ona reference stations, and only using concurrent data for March 2018 - 2019.  The wind roses are overlaid on the terrain and  highlight the terrain forcing at each site. As only concurrent data is used, wind conditions at all masts can be compared to each other. The text has been changed accordingly and improved considerably (fourth paragraph of section 4).**

2. Quality control is an important part of data description. I think that Section3 Data handling and Quality should be dedicated to quality control. The current content is mainly on processing and transmission, that seems to fit in Section 5. What is the latency of the data—near real time?

**Response: The data are made available on thredds at MET for potential users to use as fit. However, beyond the filtering already done, no additional filtering of the dataset can be done in connection with producing this manuscript.  That is the data is provided as is and a more filtering is beyond the scope of this paper and the budget for making the data available.  We believe that this does not render the dataset less valuable for the potential user, and that the data is still useful and a description of it**

**still should be published, in order to facilitate the use of the data. However, this means the user must put an effort into the filtering routines he sees needed for his intended use of the dataset. The latency of the data is on the order of hours to days, as now stated in the manuscript.**

**Changes: Section 3 has been expanded and improved, with more details on the quality control which has been employed on the 10 Hz data. Some parts of the previous section 3 were moved to section 5.**

3. Figure 13: How long does the observation record go back at the reference site? Is it available before 2009? If so, it would be nice to use longer time mean.

**Response: ONA II has been operational since 1978 and we have used 18 years of data (2001-2019) with hourly temporal resolution. Before 2001, recordings were less frequent, not automated and done with a different instrument. We have therefore chosen to use the longest homogeneous part of the series in order to not introduce any discrepancies which could be associated with a longer time series.**

**Changes: We have modified the first paragraph of section 4.**

4. Line 246-247: I don't understand this sentence, please clarify.

**Response: We agree that the sentence was unclear.**

**Changes: We have rephrased the sentence to better convey the message, and it now reads:** "When compared to the wind speed distribution for the reference period of 18 years (Fig. 9) we see that the wind has been somewhat weaker during the last 3 years than during the reference period."

**Response to these comments were already published, and we have taken them into account when revising the manuscript as indicated below.**

*The manuscript "Meteorological observations in tall masts for mapping of atmospheric flow in Norwegian fjords" by Furevik et al. deals with a unique data set, freely accessible since 2018, which is particularly valuable for both engineers and scientists working on the E39 Coastal Highway Route (Ferry Free E39). Nevertheless, some of the statements in the manuscript may be unclear, ambiguous or misleading:*

*• Line 59, the authors mention that the "dataset provides invaluable data describing the atmospheric forcing, both climatic and short-term, pertaining to the technical design of large structures in complex terrain." Although I understand the enthusiasm of the authors, one should keep in mind that the potential and limits of the dataset have not yet been assessed in details.*

**Changes: The sentence is rephrased and some text is added in paragraph 4 of section 1.**

*It is also unclear to me what the authors mean exactly by "atmospheric forcing, both climatic and short-term" with respect to structural design. A more specific reformulation would be welcome.*

**Changes: We have changed the word to "forces".**

*• Line 34-35. As the authors already know, there has been a similar campaign in the Bjørnafjord since 2015. Although the data in that fjord are not publicly available, it may be useful to the reader to know that the campaigns in the Sulafjord, Halsafjord and Julsundet are not the only ones.*

**Changes: We have added a sentence in paragraph 2 of section 1 about the other measurement campaigns of NPRA.**

*• Line 98: If no filtering is applied beforehand, downsampling a time series from 20 Hz to 10 Hz will amplify aliasing not reduce it. In general, downsampling increases aliasing. As far as I know, the downsampling procedure was done without filtering, resulting in undesirable aliasing, visible in Figure 15, at frequencies above 4 Hz.*

**Changes: No change was made since the temporal resolution is decreased by averaging two samples.**

• It may be informative to the reader to know if the high-frequency sonic temperature is freely available or not. I am aware that some 2-Hz sonic temperature records are usable, but this sampling frequency may be too low to study turbulent fluxes. A sampling frequency of 10 Hz or more is desirable for such purposes.

**Changes: We have added a sentence at the end of paragraph 2 in section 2.1.**

[revised manuscript text omitted]

---

## Author Response (AR2)

Dear editor/reviewer

Thank you for the decision and the final comments to correct. We have revised the manuscript according to the comments and changed a few minor errors. The associated change to the manuscript is described below each point and a marked-up version of the document follows.

*1) change "done" to "made" in Line#12.*

This is corrected.

*2) avoid the terms "climate" or "climatological". (The length of your records is well below the classical "threshold" (30 years) defined by the World Meteorological Organization. It is better to use "18-year reference period" than "18-year climatological period".)*

This has been changed everywhere in the manuscript.

*3) clarify the choice to present a (simple) time series instead of vertical wind profiles. (I don't understand why the authors did not produce a figure showing an example of wind profiles, which, in my opinion, can potentially represent the real "added-value" of the presented dataset.)*

We have decided to include an example of vertical wind profiles in a new figure and added text in line 296-302 and 324-327 to explain.

[revised manuscript text omitted]